# Conformal Bayesian Computation

**Edwin Fong**
University of Oxford
The Alan Turing Institute
edwin.fong@stats.ox.ac.uk

**Chris Holmes**
University of Oxford
The Alan Turing Institute
cholmes@stats.ox.ac.uk

## Abstract

We develop scalable methods for producing conformal Bayesian predictive intervals with finite sample calibration guarantees. Bayesian posterior predictive distributions, $p(y \mid x)$, characterize subjective beliefs on outcomes of interest, $y$, conditional on predictors, $x$. Bayesian prediction is well-calibrated when the model is true, but the predictive intervals may exhibit poor empirical coverage when the model is misspecified, under the so called $\mathcal{M}$-open perspective. In contrast, conformal inference provides finite sample frequentist guarantees on predictive confidence intervals without the requirement of model fidelity. Using 'add-one-in' importance sampling, we show that conformal Bayesian predictive intervals are efficiently obtained from re-weighted posterior samples of model parameters. Our approach contrasts with existing conformal methods that require expensive refitting of models or data-splitting to achieve computational efficiency. We demonstrate the utility on a range of examples including extensions to partially exchangeable settings such as hierarchical models.

## 1   Introduction

We consider Bayesian prediction using training data $Z_{1:n} = \{Y_i, X_i\}_{i=1:n}$ for an outcome of interest $Y_i$ and covariates $X_i \in \mathbb{R}^d$. Given a model likelihood $f_\theta(y \mid x)$ and prior on parameters, $\pi(\theta)$ for $\theta \in \mathbb{R}^p$, the posterior predictive distribution for the response at a new $X_{n+1} = x_{n+1}$ takes on the form

$$p(y \mid x_{n+1}, Z_{1:n}) = \int f_\theta(y \mid x_{n+1}) \, \pi(\theta \mid Z_{1:n}) \, d\theta \,, \tag{1}$$

where $\pi(\theta \mid Z_{1:n})$ is the Bayesian posterior. Asymptotically exact samples from the posterior can be obtained through Markov chain Monte Carlo (MCMC) and the above density can be computed through Monte Carlo (MC), or by direct sampling from an approximate model. Given a Bayesian predictive distribution, one can then construct the highest density $100 \times (1 - \alpha)\%$ posterior predictive credible intervals, which are the shortest intervals to contain $(1 - \alpha)$ of the predictive probability. Alternatively, the central $100 \times (1 - \alpha)\%$ credible interval can be computed using the $\alpha/2$ and $1 - \alpha/2$ quantiles. Posterior predictive distributions condition on the observed $Z_{1:n}$ and represent subjective and coherent beliefs. However, it is well known that model misspecification can lead Bayesian intervals to be poorly *calibrated* in the frequentist sense (Dawid, 1982; Fraser et al., 2011), that is the long run proportion of the observed data lying in the $(1 - \alpha)$ Bayes predictive interval is not necessarily equal to $(1 - \alpha)$. This has consequences for the robustness of such approaches and trust in using Bayesian models to aid decisions.

Alternatively, one can seek intervals around a point prediction from the model, $\widehat{y} = \widehat{\mu}(x)$, that have the correct frequentist coverage of $(1 - \alpha)$ . This is precisely what is offered by the *conformal prediction* framework of Vovk et al. (2005), which allows the construction of prediction bands with finite sample validity without assumptions on the generative model beyond exchangeability of the

35th Conference on Neural Information Processing Systems (NeurIPS 2021).

data. Formally, for $Z_i = \{Y_i, X_i\}_{i=1:n}$, $Z_i \overset{\text{iid}}{\sim} \mathbb{P}$ and miscoverage level $\alpha$, conformal inference allows us to construct a confidence set $C_\alpha(X_{n+1})$ from $Z_{1:n}$ and $X_{n+1}$ such that

$$\mathbb{P}(Y_{n+1} \in C_\alpha(X_{n+1})) \geq 1 - \alpha. \tag{2}$$

Note that $\mathbb{P}$ is over $Z_{1:n+1}$, that is we also treat $X_{n+1}$ as random; see Lei and Wasserman (2014) for a discussion on the difficulties of attaining coverage conditional on $X_{n+1}$. In this paper we develop computationally efficient conformal inference methods for Bayesian models including extensions to hierarchical settings. A general theme of our work is that, somewhat counter-intuitively, Bayesian models are well suited for the conformal method.

Conformal inference for calibrating Bayesian models was previously suggested in Melluish et al. (2001), Vovk et al. (2005), Wasserman (2011) and Burnaev and Vovk (2014), where it is referred to as "de-Bayesing", "frequentizing" and "conformalizing", but only in the context of conjugate models. Here, we present a scalable MC method for *conformal Bayes*, implementing full conformal Bayesian prediction using an 'add-one-in' importance sampling algorithm. The automated method can construct conformal predictive intervals from any Bayesian model given only samples of model parameter values from the posterior $\theta \sim \pi(\theta \mid Z_{1:n})$, up to MC error. Such samples are readily available in most Bayesian analyses from probabilistic programming languages such as Stan (Carpenter et al., 2017) and PyMC3 (Salvatier et al., 2016). We also extend conformal inference to partially exchangeable settings which utilize the important class of Bayesian hierarchical models, and note the connection to Mondrian conformal prediction (Vovk et al., 2005, Chapter 4.5). Previously, the extension of conformal prediction to random effects was introduced in Dunn et al. (2020) in a non-Bayesian setting, with a focus on prediction in new groups, as well as within-group predictions without covariates. We will see that the Bayesian hierarchical model allows for a natural sharing of information between groups for within-group predictions with covariates. We discuss the motivation behind using the Bayesian posterior predictive density as the conformity measure for both the Bayesian and the frequentist, and demonstrate the benefits in a number of examples.

## 1.1 Background

The conformal inference framework was first introduced by Gammerman et al. (1998), followed by the thorough book of Vovk et al. (2005). Full conformal prediction is computationally expensive, requiring the whole model to be retrained at each test covariate $x_{n+1}$ *and* for each value in a reference grid of potential outcomes, e.g. $y \in \mathbb{R}$ for regression. This makes the task computationally infeasible beyond a few special cases where we can shortcut the evaluation along the outcome reference grid, e.g. ridge regression (Vovk et al., 2005; Burnaev and Vovk, 2014) and Lasso (Lei, 2019). Shrinking the search grid is possible, but still requires many refittings of the model (Chen et al., 2016). The split conformal prediction method (Lei et al., 2018) is a useful alternative method which only requires a single model fit, but increases variability by dividing the data into a training and test set that includes randomness in the choice of the split, and has a tendency for wider intervals. Methods based on cross-validation such as cross-conformal prediction (Vovk, 2015) and the jacknife+ (Barber et al., 2021) lie in between the split and full conformal method in terms of computation and average length of the intervals. A detailed comparison of various conformal methods are provided in Barber et al. (2021, Section 4). A review of recent advances in conformal prediction is given in Zeni et al. (2020), and interesting extensions have been developed by works such as Tibshirani and Foygel (2019); Romano et al. (2019); Candès et al. (2021).

## 2 Conformal Bayes

### 2.1 Full Conformal Prediction

We begin by summarizing the full conformal prediction algorithm discussed in Vovk et al. (2005); Lei et al. (2018). Firstly, a conformity (goodness-of-fit) measure,

$$\sigma_i := \sigma(Z_{1:n+1}; Z_i),$$

takes as input a set of data points $Z_{1:n+1}$, and computes how similar the data point $Z_i$ is for $i = 1, \ldots, n+1$. A typical conformity measure for regression would be the negative squared error arising from a point prediction $-\{y_i - \widehat{\mu}(x_i)\}^2$, where $\widehat{\mu}(x)$ is the point predictor fit to the augmented dataset $Z_{1:n+1}$, assumed to be symmetric with respect to the permutation of the input

dataset. The key property of any conformity measure is that it is exchangeable in the first argument, i.e. the conformity measure for $Z_i$ is invariant to the permutation of $Z_{1:n+1}$. Under the assumption that $Z_{1:n+1}$ is exchangeable, we then have that $\sigma_{1:n+1}$ is also exchangeable, and its rank is uniform among $\{1, \ldots, n+1\}$ (assuming continuous $\sigma_{1:n+1}$). From this, we have that the rank of $\sigma_{n+1}$ is a valid $p$-value. If we now consider a plug-in value $Y_{n+1} = y$ (where $X_{n+1}$ is known), we can define the rank of $\sigma_{n+1}$ among $\sigma_{1:n+1}$ as

$$r(y) = \frac{1}{n+1} \sum_{i=1}^{n+1} \mathbb{1}\left(\sigma_i \leq \sigma_{n+1}\right).$$

For miscoverage level $\alpha$, the full conformal predictive set,

$$C_\alpha(X_{n+1}) = \{y \in \mathbb{R} : r(y) > \alpha\}, \tag{3}$$

satisfies the desired frequentist coverage as in (2) under the exchangeability assumptions above. Intuitively, we are reporting the values of $y$ which conform better than the fraction $\alpha$ of observed conformity scores in the augmented dataset. A formal proof can be found in Vovk et al. (2005, Chapter 8.7). For continuous $\sigma_{1:n+1}$, we also have from Lei et al. (2018, Theorem 1) that the conformal predictive set does not significantly over-cover.

In practice, beyond a few exceptions, the function $r(y)$ must be computed on a fine grid $y \in \mathcal{Y}_{\text{grid}}$, for example of size 100, in which case the model must be retrained 100 times to the augmented dataset to compute $\sigma_{1:n+1}$, with plug-in values for $y_{n+1}$ on the grid. This is illustrated in Algorithm 1 below. We note here that the grid method only provides approximate coverage, as $y$ may be selected even if it lies between two grid points that are not selected. This is formalized in Chen et al. (2018), but we do not discuss this further. In Appendix D.6, we provide an empirical comparison of the grid effects. This is also valid for binary classification where we now have a finite $\mathcal{Y}_{\text{grid}} = \{0, 1\}$, and so the grid method for full conformal prediction is exact and feasible.

---

Observed data is $Z_{1:n}, X_{n+1}$; Specify miscoverage level $\alpha$
**for** each $y \in \mathcal{Y}_{\text{grid}}$ **do**
    Fit model to augmented dataset $\{Z_1, \ldots, Z_n, \{y, X_{n+1}\}\}$
    Compute $\sigma_{1:n}$ and $\sigma_{n+1}$
    Store the rank, $r(y)$, of $\sigma_{n+1}$ among $\sigma_{1:n+1}$
**end**
Return the set $C_\alpha(X_{n+1}) = \{y \in \mathcal{Y}_{\text{grid}} : r(y) > \alpha\}$.

**Algorithm 1:** Full Conformal Prediction

## 2.2 Conformal Bayes and Add-One-In Importance Sampling

In a Bayesian model, a natural suggestion for the conformity score, as noted in Vovk et al. (2005); Wasserman (2011), is the posterior predictive density (1), that is

$$\sigma(Z_{1:n+1}; Z_i) = p(Y_i \mid X_i, Z_{1:n+1})$$

for $i \in \{1 \ldots, n+1\}$, where

$$p(Y_i \mid X_i, Z_{1:n+1}) = \int f_\theta(Y_i \mid X_i)\, \pi(\theta \mid Z_{1:n+1})\, d\theta. \tag{4}$$

We highlight that the datum $Z_i = \{Y_i, X_i\}$ also lies in the conditioning $Z_{1:n+1}$. This is a valid conformity score, as we have $\pi(\theta \mid Z_{1:n+1}) \propto \pi(\theta) \prod_{i=1}^{n+1} f_\theta(Y_i \mid X_i)$, and so $\sigma$ is indeed invariant to the permutation of $Z_{1:n+1}$. We refer to this method as *conformal Bayes* (CB) - this is a special case of full conformal prediction, and we will see shortly that the exchangeability structure of Bayesian models is key to constructing conformity scores in the partial exchangeability scenario.

Beyond conjugate models, we are usually able to obtain (asymptotically exact) posterior samples $\theta^{(1:T)} \sim \pi(\theta \mid Z_{1:n})$, e.g through MCMC, where $T$ is a large integer. Such samples are typically available as standard output from Bayesian model fitting. The posterior predictive can then be computed up to Monte Carlo error through

$$\widehat{p}(Y_{n+1} \mid X_{n+1}, Z_{1:n}) = \frac{1}{T} \sum_{t=1}^{T} f_{\theta^{(t)}}(Y_{n+1} \mid X_{n+1}).$$

The key insight is that refitting the Bayesian model with $\{Z_1, \ldots, Z_n, \{y, X_{n+1}\}\}$ is well approximated through importance sampling (IS), as only $\{y, X_{n+1}\}$ changes between refits. This leads immediately to an IS approach to full conformal Bayes, where we just need to compute 'add-one-in' (AOI) predictive densities. Here AOI refers to the inclusion of $\{Y_{n+1}, X_{n+1}\}$ into the training set, named in relation to 'leave-one-out' (LOO) cross-validation. Specifically, for $Y_{n+1} = y$ and $\theta^{(1:T)} \sim \pi(\theta \mid Z_{1:n})$, we can compute

$$\widehat{p}(Y_i \mid X_i, Z_{1:n+1}) = \sum_{t=1}^{T} \widetilde{w}^{(t)} f_{\theta^{(t)}}(Y_i \mid X_i) \tag{5}$$

where $\widetilde{w}^{(t)}$ are our self-normalized importance weights of the form

$$w^{(t)} = f_{\theta^{(t)}}(y \mid X_{n+1}), \quad \widetilde{w}^{(t)} = \frac{w^{(t)}}{\sum_{t'=1}^{T} w^{(t')}}. \tag{6}$$

We see that the unnormalized importance weights have the intuitive form of the predictive likelihood at the reference point $\{y, X_{n+1}\}$ given the model parameters $\theta^{(t)}$.

The use of AOI importance sampling has similarities to the computation of Bayesian leave-one-out (LOO) predictive densities for cross-validation (Vehtari et al., 2017), which is also used in accounting for model misspecification. An interesting aspect of AOI in comparison with LOO is that AOI predictive densities are less vulnerable to importance weight instability for the following reasons:

- In LOO, the target $\pi(\theta \mid Z_{-i})$ generally has thicker tails than the proposal $\pi(\theta \mid Z_{1:n})$, leading to importance weight instability. In contrast, AOI uses the posterior $\pi(\theta \mid Z_{1:n})$ as a proposal for the thinner-tailed $\pi(\theta \mid Z_{1:n+1})$. For LOO the importance weights are proportional to $1/f_\theta(y \mid x)$, in contrast to the typically bounded $f_\theta(y \mid x)$ for AOI.
- For AOI, we are predicting $Z_i$ given $Z_{1:n+1}$ which is always in-sample unlike in LOO where the datum is out-of-sample, so we can expect greater stability with AOI.
- The IS weight stability is governed by $Y_{n+1} = y$, which is not random as we select it for the grid, unlike in LOO cross-validation. For sufficiently large $\alpha$, we will not need to compute the AOI predictive density for extreme values of $y$.

The above discussion is a natural analogy to the discussion provided in Vehtari et al. (2017, Section 2.1.1) for computing LOO predictives. Note that the LOO predictive density is also a valid conformity score; we provide more details and verify the above comparisons between LOO and AOI empirically in Appendix B. We also provide some IS weight diagnostics in Section 4.1.1 and find that they are stable. In difficult settings such as very high-dimensions, one can make use of the recommendations of Vehtari et al. (2015) for assessing and Pareto-smoothing the importance weights if necessary.

## 2.3 Computational Complexity

Given the posterior samples, we must compute the likelihood for each $\theta^{(t)}$ at $Z_{1:n}$, as well at $\{y, X_{n+1}\}$ for $y \in \mathcal{Y}_{\text{grid}}$. The additional computation required for CB for each $X_{n+1}$ is thus $T \times (n + n_{\text{grid}})$ likelihood evaluations, which is relatively cheap. This is then followed by the dot product of an $(n + 1) \times T$ matrix with a $T$ vector for each $y$, which is $\mathcal{O}(nT)$, so the overall complexity is $\mathcal{O}(n_{\text{grid}}Tn)$. The large matrices involved in computing the AOI predictives suggests we can take advantage of GPU computation, and machine learning packages such as JAX (Bradbury et al., 2018) are highly suitable for this application.

The values $n_{\text{grid}}$ and $T$ are constants we select, and we now briefly provide guidance on this. Selecting a grid is a general problem for full conformal prediction (beyond just CB), and this has been investigated in Chen et al. (2016, 2018). Here, we opt for a default choice of $n_{\text{grid}} = 100$ in Section 4.1 and we show in Appendix D.6 that this is sufficient for accurate coverage. For the selection of $T$, there are standard methods based on the effective sample size (ESS) which we compute in Section 4.1.1. One could then approximate the Monte Carlo standard error in the density estimates which is proportional to $1/\sqrt{\text{ESS}}$, or estimate the IS variance directly. The ESS will depend both on MCMC mixing and IS stability, which may further depend on the dimensionality of the model. More details can be found in Owen (2013, Chapter 9.3)

## 2.4 Motivation

Much has been written on the contrasting foundations and interpretation of Bayes versus frequentist measures of uncertainty (Little, 2006; Shafer and Vovk, 2008; Bernardo and Smith, 2009; Wasserman, 2011), and we provide a summary in Appendix A. Here we motivate CB predictive intervals from both a Bayesian and frequentist perspective.

The pragmatic Bayesian, aware of the potential for model misspecification in either the prior or likelihood, may be interested in conformal inference as a countermeasure. CB predictive intervals with guaranteed frequentist coverage can be provided as a supplement to the usual Bayesian predictive intervals. The difference between the Bayesian and conformal interval may also serve as an informal diagnostic for model evaluation (e.g. Gelman et al. (2013)). Posterior samples through MCMC or direct sampling are typically available, and so CB through automated AOI carries little overhead.

The frequentist may also wish to use a Bayesian model as a tool for constructing predictive confidence intervals. Firstly, the likelihood can take into account skewness, heteroscedasticity unlike the usual residual conformity score. Secondly, features such as sparsity, support, and regularization can be included through priors, while CB ensures correct coverage. Thirdly, CB provides a computationally feasible method to compute full conformal prediction intervals for a larger class of models. Finally, a subtle issue that arises in full conformal prediction is that we lose validity if hyperparameter selection is not symmetric with respect to $Z_{n+1}$, e.g. if we estimate the Lasso penalty $\lambda$ using only $Z_{1:n}$ before computing the full conformal intervals with said $\lambda(Z_{1:n})$. For CB, a prior on hyperparameters induces weighting of the hyperparameter values by implicit cross-validation for each refit (Gneiting and Raftery, 2007; Fong and Holmes, 2020). We highlight here that this issue does not affect the split conformal method, e.g. Lei and Wasserman (2014, Section 5).

## 3 Partial Exchangeability and Hierarchical Models

A setting of particular interest is for grouped data, which corresponds to a weakening of exchangeability often referred to as partial exchangeability (Bernardo and Smith, 2009, Chapter 4.6). Assume that we observe data from $J$ groups, each of size $n_j$, where again $Z_{i,j} = \{Y_{i,j}, X_{i,j}\}$. We write the full dataset as $Z = \{Z_{i,j} : i = 1, \ldots, n_j, j = 1, \ldots, J\}$. We may not expect the entire sequence $Z$ to be exchangeable, instead only that data points are exchangeable within groups. Formally, this means that

$$p(Z_{1:n_1,1}, \ldots, Z_{1:n_J,J}) = p(Z_{\gamma_1(1):\gamma_1(n_1),1}, \ldots, Z_{\gamma_J(1):\gamma_J(n_J),J}) \tag{7}$$

for any permutations $\gamma_j$ of $1, \ldots, n_j$, for $j = 1, \ldots, J$. Alternatively, we can enforce the usual definition of exchangeability but only consider permutations of $1, \ldots, n$ such that the groupings are preserved. A simple example of this partial exchangeability is if $Z_{i,j} \stackrel{\text{iid}}{\sim} P_j$ for $i = 1, \ldots, n_j$, $j = 1, \ldots, J$, where $P_j$ can now be distinct.

Partial exchangeability is useful in multilevel modelling, e.g. where $Z_{1:n_j,j}$ records exam results on students within school $j$, for schools $j = 1, \ldots, J$. Students may be deemed exchangeable within schools, but not between schools. Further examples may be found in Gelman and Hill (2006).

### 3.1 Group Conformal Prediction

Given a new $X_{n_j+1,j}$ belonging to group $j$ for $j \in \{1, \ldots, J\}$, we seek to construct a $(1 - \alpha_j)$ confidence interval for $Y_{n_j+1,j}$. We define a within-group conformity score as

$$\sigma_{i,j} := \sigma_{Z_{-j}}(Z_{1:n_j+1,j}; Z_{i,j})$$

for $i = 1, \ldots, n_j + 1$. We write $Z_{-j}$ as the dataset without group $j$, and $Z_{-j}$ in the subscript indicates the dependence of the conformity score on this, which we motivate in the next subsection. For each $Z_{-j}$, we require the score to be invariant with respect to the permutation of $Z_{1:n_j+1,j}$. For $Z_{n_j+1,j} = \{y, X_{n_j+1,j}\}$, the conformal predictive set is then defined

$$r_j(y) = \frac{1}{n_j + 1} \sum_{i=1}^{n_j+1} \mathbb{1}\left(\sigma_{i,j} \leq \sigma_{n_j+1,j}\right), \quad C_{\alpha_j}\left(X_{n_j+1,j}\right) = \{y \in \mathbb{R} : r_j(y) > \alpha_j\} \tag{8}$$

In other words, we rank the conformity scores $\sigma_{1:n_j+1,j}$ within the group $j$, and compute the conformal interval as usual with Algorithm 1. The interval is valid from the following.

**Proposition 1.** *Assume that $\{Z, Z_{n_j+1,j}\}$ is partially exchangeable as in (7), and the conformity measure $\sigma_{i,j}$ for group $j$ is invariant to the permutation of $Z_{1:n_j+1,j}$. We then have*

$$\mathbb{P}\left(Y_{n_j+1,j} \in C_{\alpha_j}\left(X_{n_j+1,j}\right)\right) \geq 1 - \alpha_j$$

*where $C_{\alpha_j}\left(X_{n_j+1,j}\right)$ is defined in (8), and $\mathbb{P}$ is over $\{Z, Z_{n_j+1,j}\}$.*

*Proof.* Conditional on $Z_{-j}$, the observations $Z_{1:n_j+1,j}$ are still exchangeable, and thus so are $\sigma_{1:n_j+1,j}$ from the invariance of the conformity measure. The usual conformal guarantee then holds:

$$\mathbb{P}\left(Y_{n_j+1,j} \in C_{\alpha_j}\left(X_{n_j+1,j}\right) \mid Z_{-j}\right) \geq 1 - \alpha_j.$$

Taking the expectation with respect to $Z_{-j}$ gives us the result. $\qquad\square$

It is interesting to note that the above group conformal predictor coincides with the attribute-conditional Mondrian conformal predictor of Vovk et al. (2005, Chapter 4.5), with the group allocations as the taxonomy. Validity under the relaxed Mondrian-exchangeability of Vovk et al. (2005, Chapter 8.4) is key for us here.

### 3.2 Conformal Hierarchical Bayes

Under this setting, a hierarchical Bayesian model can be defined of the form

$$[Y_{i,j} \mid X_{i,j}, \theta_j, \tau] \overset{\text{iid}}{\sim} f_{\theta_j,\tau}(\cdot \mid X_{i,j}) \quad i = 1, \dots, n_j, \qquad j = 1, \dots, J$$
$$[\theta_j \mid \phi] \overset{\text{iid}}{\sim} \pi(\cdot \mid \phi) \qquad\qquad\qquad j = 1, \dots, J$$
$$\phi \sim \pi(\phi), \quad \tau \sim \pi(\tau).$$

Here $\tau$ is a common parameter across groups (e.g. a common standard deviation for the residuals under homoscedastic errors). The desired partial exchangeability structure is clearly preserved in the Bayesian model (Bernardo, 1996). De Finetti representation theorems are also available for partially exchangeable sequences (when defined in a slightly different manner to the above), which motivate the specification of hierarchical Bayesian models (Bernardo and Smith, 2009, Chapter 4.6).

The posterior predictive is once again a natural choice for the conformity measure. Denoting $\bar{Z}_y$ as the entire dataset augmented with $Z_{n_j+1,j} = \{y, X_{n_j+1,j}\}$, we have

$$\sigma_{i,j} = p(Y_{i,j} \mid X_{i,j}, \bar{Z}_y) = \int f_{\theta_j,\tau}(Y_{i,j} \mid X_{i,j}) \, \pi(\theta_j, \tau \mid \bar{Z}_y) \, d\theta_j \, d\tau \tag{9}$$

for $i = 1, \dots, n_j + 1$. The within-group permutation invariance follows as the likelihood is exchangeable within groups, and thus so is the posterior and resulting posterior predictive. Practically, this structure allows for independent coefficients $\theta_j$ for each group, but partial pooling through $\pi(\theta \mid \phi)$ allows information to be shared between groups. A fully pooled model, whilst still valid, is usually too simple and predicts poorly, whereas a no-pooling conformity score ignores information sharing between groups. More details on hierarchical models can be found in Gelman et al. (2013, Chapter 5). We point out that we can select a separate coverage level $\alpha_j$ for each group, which will be useful when group sizes $n_j$ vary - we provide a demonstration of this in Appendix D.4.2. Computation of $\sigma_{i,j}$ is again straightforward, where MCMC now returns $[\theta_{1:J}^{(1:T)}, \phi^{(1:T)}, \tau^{(1:T)}] \sim \pi(\theta_{1:J}, \phi, \tau \mid Z)$. We can then estimate (9) using AOI importance sampling as in (5) and (6) using the marginal samples $\{\theta_j^{(1:T)}, \tau^{(1:T)}\} \sim \pi(\theta_j, \tau \mid Z)$ and weights $w^{(t)} = f_{\theta_j^{(t)}, \tau^{(t)}}(y \mid X_{n_j+1,j})$.

In the above, we consider predictive intervals within groups with covariates, extending the within-group approach of Dunn et al. (2020). Predictive intervals for new groups are possible with the Bayesian model, but a conformal predictor would require additional stronger assumptions of exchangeability to ensure validity. The relevant assumptions and methods based on pooling and subsampling for new group predictions are discussed in Dunn et al. (2020), but would require rerunning MCMC in our case. We leave this for future work, noting that utilizing the Bayesian predictive density directly here seems nontrivial due to different group sizes.

# 4   Experiments

We run and time all examples on an Azure NC6 Virtual Machine, which has 6 Intel Xeon E5-2690 v3 vCPUs and a one-half Tesla K80 GPU card. We use PyMC3 (Salvatier et al., 2016) for MCMC and `sklearn` (Pedregosa et al., 2011) for the regular conformal predictor; both are run on the CPU. Computation of the CB and Bayes intervals is implemented in JAX (Bradbury et al., 2018), and run on the GPU. The code is available online[1] and further examples are provided in Appendix D.

## 4.1   Sparse Regression

We first demonstrate our method under a sparse linear regression model on the diabetes dataset (Efron et al., 2004) considered by Lei (2019). The dataset is available in `sklearn`, and consists of $n = 442$ subjects, where the response variable is a continuous diabetes progression and the $d = 10$ covariates consist of patient readings such as blood serum measurements. We standardize all covariates and the response to have mean 0 and standard deviation 1.

The Bayesian model we consider is

$$f_\theta(y \mid x) = \mathcal{N}(y \mid \theta^{\mathrm{T}} x + \theta_0, \tau^2)$$
$$\pi(\theta_j) = \mathrm{Laplace}(0, b), \quad \pi(\theta_0) \propto 1, \quad \pi(b) = \mathrm{Gamma}(1, 1) \quad \pi(\tau) = \mathcal{N}^+(0, c) \tag{10}$$

for $j = 1, \ldots, d$, and where $b$ is the scale parameter and $\mathcal{N}^+$ is the half-normal distribution. Note that a hyperprior on $b$ has removed the need for cross-validation that is required for Lasso. We consider two values of $c$ for the hyperprior on $\tau$, which correspond to a well-specified ($c = 1$) and poorly-specified ($c = 0.02$) prior; in the latter case our posterior on $\tau$ will be heavily weighted towards a small value. This model is well-specified for the diabetes dataset (Jansen, 2013, Chapter 4.5) under a reasonable prior ($c = 1$). We compute the central $(1 - \alpha)$ credible interval from the Bayesian posterior predictive CDF estimated using Monte Carlo and the same grid as for CB.

To check coverage, we repeatedly divide into a training and test dataset for 50 repeats, with 30% of the dataset in the test split. We evaluate the conformal prediction set on a grid of size $n_{\mathrm{grid}} = 100$ between $[y_{\min} - 2, y_{\max} + 2]$, where $y_{\min}, y_{\max}$ is computed from each training dataset. The average coverage, length and run-times (*excluding* MCMC) with standard errors are given in Table 1 for $\alpha = 0.2$. MCMC induced an average overhead of 21.9s for $a = 1$ and 26.8s for $c = 0.02$ for the Bayes and CB interval, where we simulate $T = 8000$ posterior samples. The CB intervals are only slightly slower than the Bayes intervals, and still a small fraction of the time required for MCMC, and is thus an efficient post-processing step. For $c = 1$, the Bayesian intervals have coverage close to $(1 - \alpha)$ with the smallest expected length, with CB slightly wider and more conservative. However, when the prior is misspecified with $c = 0.02$, the Bayes intervals severely undercover, whilst the CB coverage and length remain unchanged from the $c = 1$ case.

As baselines, we compare to the split and full conformal method using the non-Bayesian Lasso as the predictor, with the usual residual as the nonconformity score. For the split method, we fit Lasso with cross-validation on the subset of size $n_{\mathrm{train}}/2$ to obtain the Lasso penalty $\lambda$. For the full conformal method, we use the grid method for fair timing, as other estimators beyond Lasso would not have the shortcut of Lei (2019). As setting a default $\lambda = 1$ gives poor average lengths, we estimate $\lambda = 0.004$ on cross-validation on one of the training sets, and use this value over the 50 repeats. However, we must emphasize again that this is somewhat misleading, as discussed in Section 2.4. A fairer approach would involve fitting Lasso with CV for each of the 100 grid values and 133 test values, but this is infeasible as each fit requires around 80ms, resulting in a total run-time of 17 minutes. On the other hand, the AOI scheme of CB is equivalent to refitting $b$ for each grid/test value. In terms of performance, the split method has wider intervals than CB/full, but performs well given the low computational costs. The full conformal method performs as well as CB, but is comparable in time as MCMC + CB, whilst not refitting $\lambda$. We note that the value of $c$ does not affect the split/full method.

---

[1]`https://github.com/edfong/conformal_bayes`

Table 1: Diabetes: Coverage values *not* within 3 standard errors (in brackets) of the target coverage $(1-\alpha) = 0.8$ are in **red**. Run-times *exclude* overhead MCMC times, which are provided in the main text.

|  |  | Bayes | CB | Split | Full ($\lambda = 0.004$) |
|---|---|---|---|---|---|
| Coverage | $c = 1$ | 0.806 (0.005) | 0.808 (0.006) | 0.809 (0.006) | 0.808 (0.006) |
|  | $c = 0.02$ | **0.563** (0.006) | 0.809 (0.006) | / | / |
| Length | $c = 1$ | 1.84 (0.01) | 1.87 (0.01) | 1.91 (0.02) | 1.86 (0.01) |
|  | $c = 0.02$ | 1.14 (0.00) | 1.87 (0.01) | / | / |
| Run-time | $c = 1$ | 0.488 (0.107) | 0.702 (0.019) | 0.065 (0.001) | 11.529 (0.232) |
| (secs) | $c = 0.02$ | 0.373 (0.002) | 0.668 (0.003) | / | / |

### 4.1.1 Importance Weights

For the diabetes dataset, we look at the ESS of the self-normalized importance weights (6), which can be computed as $\text{ESS} = 1/\sum_{t=1}^{T}\{w^{(t)}\}^2$ for each $x_{n+1}$ and $y$. The ESS as a function of $y$ for a single $x_{n+1}$ is shown in Figure 1 for the two cases $c = 1, 0.02$, with the CB conformal bands given for $\alpha = 0.2, 0.5$. We have scaled the ESS plots by $\text{ESS}_{\text{MCMC}}/T$, where $T = 8000$ is the number of posterior samples and $\text{ESS}_{\text{MCMC}}$ is the minimum ESS out of all posterior parameters return by PyMC3. We observe the ESS is well behaved and stable across the range of $y$ values. In both cases, the ESS for $\alpha = 0.2$ is sufficiently large for a reliable estimate of the conformity scores. However, for $c = 0.02$, the ESS decays more quickly with $y$ as the Bayes predictive intervals are too narrow, which the CB corrects for. Other values of $x_{n+1}$ produce similar behaviour.

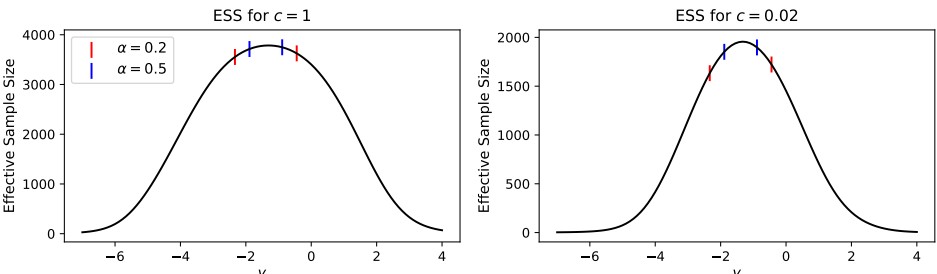

Figure 1: Effective sample sizes of IS weights with CB conformal bands for diabetes dataset with (left) $c = 1$ and (right) $c = 0.02$.

## 4.2 Sparse Classification

In this section, we analyze the Wisconsin breast cancer (Wolberg and Mangasarian, 1990), again available in `sklearn`. The dataset is of size 569, where the binary response variable corresponds to a malignant or benign tumour. The 30 covariates consist of measurements of cell nuclei. Again, we standardize all covariates to have mean 0 and standard deviation 1.

We consider the logistic likelihood $f_\theta(y = 1 \mid x) = [1 + \exp\{-(\theta^\mathsf{T} x + \theta_0)\}]^{-1}$, with the same priors for $\theta, \theta_0$ as in (10). The Bayesian predictive set is the smallest set from $\{0\}, \{1\}, \{0, 1\}$ that contains at least $(1 - \alpha)$ of the posterior predictive probability. The conformal baselines are as above but with $L_1$-penalized logistic regression, and for the full conformal method we have $\lambda = 1$. We again have 50 repeats with 70-30 train-test split, and set $\alpha = 0.2$. The grid method is now exact, and the size of the CB intervals can take on the values $\{0, 1, 2\}$. The results are provided in Table 2, where MCMC required an average of 45.4s to produce $T = 8000$ samples. We see that even with reasonable priors, Bayes can over-cover substantially, which CB corrects in roughly the same amount of time as it takes to compute the usual Bayes interval. However, we point out that CB may produce empty prediction sets, whereas Bayes cannot, and we investigate this in Appendix D.3.2.

Table 2: Breast Cancer: Coverage values *not* within 3 standard errors (in brackets) of the target coverage $(1 - \alpha) = 0.8$ are in **red**. Run-times *exclude* overhead MCMC times, which are provided in the main text. "Size" denotes the average number of elements in the conformal prediction set, averaged over the test points and repetitions.

|                 | Bayes          | CB             | Split          | Full           |
| --------------- | -------------- | -------------- | -------------- | -------------- |
| Coverage        | **0.990** (0.001) | 0.812 (0.005) | 0.809 (0.006) | 0.811 (0.005) |
| Size            | 1.06 (0.00)    | 0.81 (0.00)    | 0.81 (0.01)    | 0.81 (0.00)    |
| Run-time (secs) | 0.364 (0.007)  | 0.665 (0.012)  | 0.079 (0.002)  | 1.008 (0.016)  |

## 4.3 Hierarchical Model

We now demonstrate Bayesian conformal inference using a hierarchical Bayesian model for multilevel data. We stick to the varying intercept and varying slope model (Gelman et al., 2013), that is for $j = 1, \ldots, J$:

$$f_{\theta_j, \tau}(y_{i,j}) = \mathcal{N}(y_{i,j} \mid \theta_j^{\mathrm{T}} X_{i,j} + \theta_{0,j}, \tau^2)$$
$$\pi(\theta_j) = \mathcal{N}(\phi, s^2), \quad \pi(\theta_{0,j}) = \mathcal{N}(\phi_0, s_0^2) \tag{11}$$

with hyperpriors $\mathcal{N}(0, 1)$ on the location parameters $\phi, \phi_0$ and $\mathrm{Exp}(1)$ on the standard deviations $s, s_0, \tau$. We now apply this to a simulated example, and an application to the radon dataset of Gelman and Hill (2006) is given in Appendix D.4.2.

We consider two simulation scenarios, with $J = 5$ groups and $n_j = 10$ elements per group:

1. Well-specified: We generate group slopes $\theta_j \overset{\text{iid}}{\sim} \mathcal{N}(0, 1)$ for $j = 1, \ldots, J$. For each $j$, we generate $X_{i,j} \sim \mathcal{N}(0, 1)$ and $Y_{i,j} \sim \mathcal{N}(\theta_j X_{i,j}, 1)$.

2. Misspecified: We generate group slopes and variances $\theta_j \overset{\text{iid}}{\sim} \mathcal{N}(0, 1), \tau_j \overset{\text{iid}}{\sim} \mathrm{Exp}(1)$ for $j = 1, \ldots, J$. For each $j$, we generate $X_{i,j} \sim \mathcal{N}(0, 1)$ and $Y_{i,j} \sim \mathcal{N}(\theta_j X_{i,j}, \tau_j^2)$.

The first scenario has homoscedastic noise between groups as assumed in the model (11) whereas the second scenario is heteroscedastic between groups. To evaluate coverage, we only draw $\theta_{1:J}, \tau_{1:J}$ once (and not per repeat), giving us the values

$$\theta_{1:J} = [1.33, -0.77, -0.32, -0.99, -1.07], \quad \tau_{1:J} = [1.24, 2.30, 0.76, 0.28, 1.11].$$

For each of the 50 repeats, we draw $n_j = 10$ training and test data points from each group using the above $\theta_{1:J}$ (and $\tau_{1:J}$ for scenario 2), and report test coverage and lengths within each group. We use a grid of size 100 between $[-10, 10]$. The group-wise average lengths and coverage are given in Table 3 again with $\alpha = 0.2$. Again run-times are given post-MCMC, where MCMC required an average of 90.1s and 78.4s for scenarios 1 and 2 respectively to generate $T = 8000$ samples. The Bayes interval is again the central $(1 - \alpha)$ credible interval. The CB and Bayes methods have comparable run-times, likely due to the small $n$. As a reference, fitting a linear mixed-effects model in `statsmodels` (Seabold and Perktold, 2010) to the dataset takes around 200ms, so the full conformal method, which requires refitting for each of the 100 grid value and 50 test values, would take a total of 17 minutes. For scenario 1, both Bayes and CB provide close to $(1 - \alpha)$ coverage, with the Bayes lengths being smaller. This is unsurprising, as the Bayesian model is well-specified. In scenario 2, the Bayes intervals noticeably over/under-cover depending on the value of $\tau_{1:J}$ in relation to the Bayes posterior mean $\bar{\tau} \approx 1.3$. CB is robust to this, adapting its interval lengths accordingly (in particular for Groups 2 and 4) and providing within-group validity.

Table 3: Simulated grouped dataset; Coverage values *not* within 3 standard errors (in brackets) of the target coverage $(1 - \alpha) = 0.8$ are in **red**. Run-times *exclude* overhead MCMC times, which are provided in the main text.

|  |  | Scenario 1 | | Scenario 2 | |
|---|---|---|---|---|---|
|  | Group | Bayes | CB | Bayes | CB |
| Coverage | 1 | 0.808 (0.020) | 0.794 (0.022) | 0.826 (0.020) | 0.786 (0.025) |
|  | 2 | 0.800 (0.019) | 0.812 (0.024) | **0.522** (0.027) | 0.812 (0.024) |
|  | 3 | 0.824 (0.017) | 0.824 (0.022) | **0.974** (0.008) | 0.824 (0.020) |
|  | 4 | 0.786 (0.017) | 0.798 (0.022) | **1.000** (0.000) | 0.836 (0.021) |
|  | 5 | 0.772 (0.019) | 0.810 (0.020) | 0.826 (0.022) | 0.796 (0.022) |
|  | Overall | 0.798 (0.009) | 0.808 (0.009) | **0.830** (0.010) | 0.811 (0.009) |
| Length | 1 | 2.80 (0.05) | 3.19 (0.13) | 3.65 (0.08) | 4.01 (0.17) |
|  | 2 | 2.76 (0.05) | 3.21 (0.15) | 3.61 (0.08) | 7.27 (0.33) |
|  | 3 | 2.75 (0.04) | 3.07 (0.13) | 3.59 (0.08) | 2.28 (0.09) |
|  | 4 | 2.75 (0.05) | 3.05 (0.12) | 3.57 (0.08) | 1.23 (0.04) |
|  | 5 | 2.78 (0.05) | 3.14 (0.11) | 3.61 (0.08) | 3.47 (0.12) |
|  | Overall | 2.77 (0.04) | 3.13 (0.06) | 3.61(0.08) | 3.65 (0.09) |
| Run-time (secs) | Overall | 0.222 (0.002) | 0.381 (0.009) | 0.221 (0.002) | 0.375 (0.002) |

## 5   Discussion

In this work, we have introduced the AOI importance sampling scheme for conformal Bayesian computation, which allow us to construct frequentist-valid predictive intervals from a baseline Bayesian model using the output of an MCMC sampler. This extends naturally to the partially exchangeable setting and hierarchical Bayesian models.

Under model misspecification, or the $\mathcal{M}$-open scenario (Bernardo and Smith, 2009), CB can produce calibrated intervals from the Bayesian model. In the partially exchangeable case, CB can remain valid within groups. We find that even under reasonable priors, Bayesian predictives can over-cover, and CB can help reduce the length of intervals to get closer to nominal coverage. Diagnosing Bayesian miscalibration is in general non-trivial, but CB intervals may provide a diagnostic tool while also automatically correcting the miscalibration. When posterior samples of model parameters are available, AOI importance sampling is only a minor increase in computation, and interestingly is much faster than the split method (with the Bayesian model) which would require another run of MCMC. For the frequentist, CB intervals enjoy the tightness of the full conformal method, for a single expensive fit with MCMC followed by a cheap refitting process. We are also free to incorporate prior information, and use more complex likelihoods or priors, as well as automatically fitting hyperparameters.

There are however limitations to our approach, dictated by the realities of MCMC and IS. Firstly, the intervals are only approximate up to MC error, although we expect some robustness as the CB intervals are computed from the ranks and not the exact values. The stability of AOI importance sampling also depends on the posterior predictive being a good proposal, which may break down if the addition of the new datum $\{y, X_{n+1}\}$ has very high effect on the posterior. Finally, if our Bayesian model is grossly misspecified and the predictive densities are too narrow, the IS weights may have low ESS and the AOI estimates may have high variance.

If only approximate posterior samples are available, e.g. through variational Bayes (VB), then an AOI scheme may still be feasible, where one includes an additional correction term in the IS weights for the VB approximation, e.g. in Magnusson et al. (2019). However, this remains to be investigated, as the IS weight stability is strongly reliant on the quality of the approximate posterior. Combining this with the Pareto-smoothed IS method of Vehtari et al. (2015) may lead to a feasible solution. In our experience, CB intervals tend to be a single connected interval, which may allow for computational shortcuts in adapting the search grid. It would also be interesting to pursue the theoretical connections between the Bayesian and CB intervals, in a similar light to Burnaev and Vovk (2014).

## Acknowledgements

We thank the anonymous reviewers for their feedback on our paper. Fong is funded by The Alan Turing Institute Doctoral Studentship, under the EPSRC grant EP/N510129/1. Holmes is supported by The Alan Turing Institute, the Health Data Research, U.K., the Li Ka Shing Foundation, the Medical Research Council, and the U.K. Engineering and Physical Sciences Research Council through the Bayes4Health programme grant EP/R018561/1.

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
