# Appendix: Conformal Bayesian Computation

**Edwin Fong**
University of Oxford
The Alan Turing Institute
edwin.fong@stats.ox.ac.uk

**Chris Holmes**
University of Oxford
The Alan Turing Institute
cholmes@stats.ox.ac.uk

## A   Bayesian and Frequentist Intervals

Bayesian predictive intervals are conditioned on the specific observed sequence $Z_{1:n}$ and make statements on the next value $[Y_{n+1} \mid X_{n+1}]$. Subjective Bayesian statements on predictions are non-refutable, and are in this sense unscientific, but are optimal according to decision theoretic foundations. In contrast, a conformal (frequentist) interval relates to the properties of intervals returned by the algorithm if run repeatedly across different datasets of size $n$. Frequentist statements are in principle verifiable and hence refutable. Specifically, as conformal Bayes (CB) is a special case of full conformal prediction, the CB interval, $C_\alpha$, satisfies the below.

**Theorem 1** (Vovk et al. (2005)). *Assume that $Z_{1:n+1}$ are exchangeable, and the conformity measure $\sigma_i$ is the AOI posterior predictive density as in (4) which is invariant to the permutation of $Z_{1:n+1}$. We then have that the confidence interval $C_\alpha$ constructed through Algorithm 1 satisfies*

$$\mathbb{P}(Y_{n+1} \in C_\alpha(X_{n+1})) \geq 1 - \alpha.$$

To us, in the hands of an expert analyst with careful prior elicitation, the Bayesian conditional argument is the more persuasive for posterior and predictive uncertainty. The Bayesian predictive provides statements of uncertainty conditional on what has been observed, and so decisions pertain to each specific dataset. However, to make such strong statements, the Bayesian must usually make the strict assumption of the model being well-specified. If we wish to ensure that the predictive coverage of reported intervals is calibrated on average across repeats under weaker assumptions, then the conformal intervals are much more suitable. More details contrasting probability and confidence can be found in Shafer and Vovk (2008, Section 2.2).

At the end of the day, the Bayes and frequentist answer different questions, and the common confusion arises when treating them as answering the same. As long as we are aware they are addressing different needs, we believe both solutions are informative and useful, and indeed that is our recommendation in this paper. In practice, it may be helpful to compute both the Bayesian and CB intervals and compare. Any large deviations could suggest that the Bayesian model is misspecified.

## B   Add-One-In Versus Leave-One-Out

In Section 2.2, we discussed the computation of AOI and LOO posterior predictive densities. The LOO predictive density is also a valid conformity score, and takes the form

$$\sigma(Z_{-i}; Z_i) = p(Y_i \mid X_i, Z_{-i})$$

where $Z_{-i} = \{Z_1, \ldots, Z_{i-1}, Z_{i+1}, \ldots, Z_{n+1}\}$, and

$$p(Y_i \mid X_i, Z_{-i}) = \int f_\theta(Y_i \mid X_i) \, \pi(\theta \mid Z_{-i}) \, d\theta. \tag{B.1}$$

The validity of the LOO predictive follows from the fact that the conformity score is still symmetric with respect to the bag $Z_{-i}$; see Lei et al. (2018, Remark 4). The AOI IS weight is given in (6),

35th Conference on Neural Information Processing Systems (NeurIPS 2021).

whereas the LOO IS weights take on the form

$$w_i^{(t)} = \frac{f_{\theta^{(t)}}(y \mid X_{n+1})}{f_{\theta^{(t)}}(Y_i \mid X_i)}, \quad \widetilde{w}_i^{(t)} = \frac{w_i^{(t)}}{\sum_{t'=1}^{T} w_i^{(t')}}. \tag{B.2}$$

Here, the LOO IS weight form arises as we add the datum $Z_{n+1}$ but remove the datum $Z_i$ from the posterior. The subscript $i$ on the weights indicates that a different set of $T$ IS weights is required for each $\sigma_i$ for $i = 1, \ldots, n+1$, unlike AOI which only requires a single set of $T$ IS weights. As a result, computing LOO predictive densities is slower due to the larger intermediate arrays that arise, while computing the AOI predictive densities only require an efficient matrix vector multiplication. We also expect the AOI intervals to be slightly narrower, as the predictive is fit to more data points than with LOO. In general, the AOI method is more standard in conformal prediction (Zeni et al., 2020, Section 2.2.2), but LOO can sometimes be preferred (Vovk et al., 2005, Page 28); we discuss an example of this at the end of this section.

We now empirically demonstrate the difference in the AOI and LOO CB intervals for the diabetes example of Section 4.1 with $c = 1$. Using the same MCMC samples from the posterior $\pi(\theta \mid Z_{1:n})$, we compute the AOI and LOO IS weights for a single test point $x_{n+1}$ and the grid $y \in \mathcal{Y}_{\text{grid}}$. In Figure 2, we plot the effective sample size (ESS) of the AOI and LOO weights as a function of test point $y$, as well as the $\alpha = 0.2$ CB intervals. We see that in this example, the CB interval for both AOI and LOO is identical up to grid discretization. However, the ESS for the LOO IS weights is slightly lower than AOI as expected and as discussed in Section 2.2. For this test point, computing the AOI and LOO intervals required 12ms and 307ms respectively on a local CPU (2.4 GHz 8-Core Intel Core i9-9980HK), which is a significant difference. As mentioned earlier, computing the AOI interval is an efficient matrix-vector multiplication, whereas the LOO interval requires expensive broadcasting to construct the $n_{\text{grid}} \times T \times n$ IS weight array. The empirical comparison also holds for other values of $x_{n+1}$.

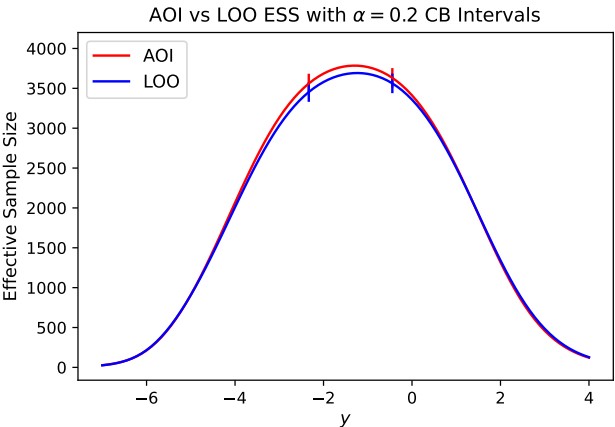

Figure 2: ESS of AOI and LOO IS weights with $\alpha = 0.2$ CB interval

We now point out an interesting example where LOO is preferred to AOI, which was suggested by an anonymous reviewer. The example given is the noiseless Gaussian process (GP), that is the model

$$Y_i = \mu(X_i), \quad \mu \sim \text{GP}(0, k(x, x')).$$

The noiseless GP interpolates between the observed data points, so the posterior predictive density $p(Y_i \mid X_i, Z_{1:n+1})$ is undefined, as $Z_i \in Z_{1:n+1}$ so the predictive variance approaches 0 as $x \to X_i$. If we assume informally that $p(Y_i \mid X_i, Z_{1:n+1}) = \infty$, then our CB interval will be the entire real line, as all conformity scores $\sigma_i$ are equal, so $r(y) = 1$ for all $y$. Note that this is not unique to CB - if we use the nonconformity score $\sigma_i = |Y_i - \mu(X_i)|$, where $\mu$ is any interpolating function such as the posterior mean of the GP, then $\sigma_i = 0$ for all $i = 1, \ldots, n+1$ and again the CB interval is the real line for the same reason. Note that this is still a valid interval, but is maximally wide. Here, the LOO predictive density (or LOO mean estimate) would not face any problems. To summarize, AOI is an inappropriate choice if the model interpolates the fitted data points $Z_{1:n+1}$.

## C  Derivation of IS weights for Hierarchical Models

We provide a quick derivation for the importance weights in Section 3.1 to estimate

$$p(Y_{i,j} \mid X_{i,j}, \bar{Z}_y) = \int f_{\theta_j, \tau}(Y_{i,j} \mid X_{i,j})\, \pi(\theta_j, \tau \mid \bar{Z}_y)\, d\theta_j\, d\tau$$

from posterior samples $[\theta_{1:J}^{(1:T)}, \tau^{(1:T)}, \phi^{(1:T)}] \sim \pi(\theta_{1:J}, \tau, \phi \mid Z)$. We can write the above as

$$p(Y_{i,j} \mid X_{i,j}, \bar{Z}_y) = \int f_{\theta_j, \tau}(Y_{i,j} \mid X_{i,j})\, \pi(\theta_{1:J}, \phi, \tau \mid \bar{Z}_y)\, d\theta_{1:J}\, d\phi\, d\tau$$

$$= \int f_{\theta_j, \tau}(Y_{i,j} \mid X_{i,j}) \frac{\pi(\theta_{1:J}, \phi, \tau \mid \bar{Z}_y)}{\pi(\theta_{1:J}, \phi, \tau \mid Z)} \pi(\theta_{1:J}, \phi, \tau \mid Z)\, d\theta_{1:J} d\phi\, d\tau$$

where

$$\frac{\pi(\theta_{1:J}, \phi, \tau \mid \bar{Z}_y)}{\pi(\theta_{1:J}, \phi, \tau \mid Z)} \propto f_{\theta_j, \tau}(y \mid X_{n_j+1, j}).$$

As the importance weight only depends on $\theta_j, \tau$, we only require the marginal posterior samples $[\theta_j^{(1:T)}, \tau^{(1:T)}]$ and we have that

$$\hat{p}(Y_{i,j} \mid X_{i,j}, \bar{Z}_y) = \sum_{t=1}^{T} w^{(t)} f_{\theta_j^{(t)}, \tau^{(t)}}(Y_{i,j} \mid X_{i,j})$$

$$w^{(t)} = f_{\theta_j^{(t)}, \tau^{(t)}}(y \mid X_{n_j+1, j}), \quad \widetilde{w}^{(t)} = \frac{w^{(t)}}{\sum_{t'=1}^{T} w^{(t')}}.$$

## D  Further Experiments

### D.1  Experimental Details

For all experiments, we repeat train-test splits or simulations 50 times, where the 70-30 train-test splits are random. For each repeat, we compute the average coverage and lengths for the test set. Means and standard errors are then computed from the 50 test set average coverages/lengths. For all MCMC examples, we generate $T = 8000$ samples, with 4000 tune steps for sparse regression/classification and 8000 for the hierarchical example.

### D.2  Sparse Regression

#### D.2.1  Diabetes

We repeat analysis on the diabetes dataset, but this time with priors

$$f_\theta(y \mid x) = \mathcal{N}(y \mid \theta^\mathsf{T} x + \theta_0, \tau^2)$$
$$\pi(\theta_j) = \text{Normal}(0, d), \quad \pi(\theta_0) = \text{Normal}(0, d), \quad \pi(\tau) = \mathcal{N}^+(0, 1) \tag{D.1}$$

where we have different values $d = 5, 0.001$ which corresponds to weak and strong regularization towards 0. For the baselines, we instead use ridge regression, with and without cross-validation for split/full as before. We emphasize that it is not exactly a fair comparison for the $d = 0.001$ case, as the baselines are tuning the parameter $\lambda$, whereas CB is subject to the misspecified prior. We still include them as baselines however, but highlight that they are not affected by the value of $d$.

MCMC required 30.8s and 13.2s for $d = 5$ and $d = 0.001$ respectively. The effect on coverage of setting $d = 0.001$ is not as detrimental as before as seen in Table 4, as the posterior on $\tau$ compensates by increasing in value; the posterior mean is $\bar{\tau} = 1$ for $d = 0.001$ versus $\bar{\tau} = 0.71$ for $d = 5$.

Table 4: Diabetes: Coverage values *not* within 3 standard errors (in brackets) of the target coverage $(1 - \alpha) = 0.8$ are in **red**. Run-times *exclude* overhead MCMC times, which are provided in the main text.

|  |  | Bayes | CB | Split | Full ($\lambda = 0.004$) |
|---|---|---|---|---|---|
| Coverage | $d = 5$ | 0.805 (0.005) | 0.809 (0.005) | 0.810 (0.006) | 0.809 (0.005) |
|  | $d = 0.001$ | **0.779** (0.006) | 0.809 (0.006) | / | / |
| Length | $d = 5$ | 1.85 (0.01) | 1.86 (0.01) | 1.91 (0.02) | 1.86 (0.01) |
|  | $d = 0.001$ | 2.56 (0.01) | 2.60 (0.01) | / | / |
| Run-time | $d = 1$ | 0.417 (0.002) | 0.677 (0.003) | 0.024 (0.000) | 8.409 (0.007) |
| (secs) | $d = 0.02$ | 0.540 (0.116) | 0.692 (0.008) | / | / |

### D.2.2  Boston Housing

The Boston housing dataset (Harrison Jr and Rubinfeld, 1978) is of size $n = 506$, consisting of $d = 13$ predictors relating to housing such as demographic and air quality, with the response as the median value of owner-occupied homes. We use the same Bayesian model as in (10), again considering $c = 1, 0.02$. For $c = 1$, the model is already misspecified for the Boston housing dataset as the errors are non-normal and have heavy tails (Jansen, 2013). All experimental settings are the same as in Section 4.1.

MCMC required an average of 22.8s and 24.4s for $c = 1, 0.02$ to produce $T = 8000$ posterior samples. Again, in Table 5 we see similar behaviour to the diabetes dataset case, but we note that even under $c = 1$, the Bayesian model over-covers. This is likely due to the presence of heavy tails in the residuals, leading to more conservative Bayesian predictive intervals. Here for $c = 1$, CB attains very close to nominal coverage and has a noticeably smaller average length. For $c = 0.02$, CB is not affected much but the Bayes interval under-covers.

Table 5: Boston: Coverage values *not* within 3 standard errors (in brackets) of the target coverage $(1 - \alpha) = 0.8$ are in **red**. Run-times *exclude* overhead MCMC times, which are provided in the main text.

|  |  | Bayes | CB | Split | Full ($\lambda = 0.004$) |
|---|---|---|---|---|---|
| Coverage | $c = 1$ | **0.860** (0.004) | 0.800 (0.005) | 0.800 (0.006) | 0.799 (0.005) |
|  | $c = 0.02$ | **0.728** (0.005) | 0.799 (0.005) | / | / |
| Length | $c = 1$ | 1.35 (0.01) | 1.12 (0.01) | 1.20 (0.02) | 1.12 (0.01) |
|  | $c = 0.02$ | 0.96 (0.00) | 1.13 (0.01) | / | / |
| Run-time | $c = 1$ | 0.414 (0.003) | 0.746 (0.011) | 0.061 (0.000) | 12.448 (0.042) |
| (secs) | $c = 0.02$ | 0.406 (0.003) | 0.744 (0.003) | / | / |

### D.3  Sparse Classification

### D.3.1  Parkinson's Disease

We provide an additional demonstration on the Parkinson's dataset (Little et al., 2008), which consists of $n = 195$ voice recordings (after removing missing data) of patients with or without Parkinson's disease encoded in the binary response. The covariates consist of $d = 22$ different voice recording properties.

The experimental setup is identical to Section 4.2, and MCMC required 29.2s to produce $T = 8000$ samples. Again, in Table 6 we see that Bayes over-covers even for reasonable priors, and CB produces tighter intervals that are closer to nominal coverage.

Table 6: Parkinson's: Coverage values *not* within 3 standard errors (in brackets) of the target coverage $(1 - \alpha) = 0.8$ are in **red**. Run-times *exclude* overhead MCMC times, which are provided in the main text. "Size" denotes the average number of elements in the conformal prediction set, averaged over the test points and repetitions.

| | Bayes | CB | Split | Full |
|---|---|---|---|---|
| Coverage | **0.955** (0.004) | 0.815 (0.008) | 0.823 (0.010) | 0.816 (0.008) |
| Size | 1.31 (0.01) | 0.93 (0.01) | 1.02 (0.02) | 0.95 (0.01) |
| Times | 0.203 (0.003) | 0.379 (0.008) | 0.478 (0.057) | 0.168 (0.003) |

### D.3.2 Uninformative Predictions

As the Bayesian model returns $p := p(y = 1 \mid x, Z_{1:n})$, we compute $(1 - \alpha)$ predictive sets by returning the smallest set of $\{0\}, \{1\}, \{0, 1\}$ such that it contains at least $(1 - \alpha)$ of the predictive probability mass. In other words, we return:

$$\begin{cases} \{0\} & \text{if } (1 - p) \geq (1 - \alpha) \\ \{1\} & \text{if } p \geq (1 - \alpha) \\ \{0, 1\} & \text{if } \max\{(1 - p), p\} \leq (1 - \alpha) \end{cases} \tag{D.2}$$

As this process is quite conservative, it is unsuprising that Bayes overcovers. The set $\{0, 1\}$ is clearly uninformative, as it is always correct. On the other hand, the conformal sets can take on the empty set $\{\}$ as well, which we know to be incorrect. As discussed in Melluish et al. (2001), empty set predictions correspond to being unable to make a prediction at the desired confidence level. Shafer and Vovk (2008) discusses the notion of *confidence* and *credibility*, which correspond to the greatest $(1 - \alpha)$ such that the conformal set is of size 1 and the greatest $\alpha$ such that the conformal set is empty respectively.

We can decompose the informative and uninformative predictive sets and look at the misclassification rate, which is the error percentage within single element predictions. In comparison, we can look at the percentage of uninformative predictives (either both elements or empty). This is shown in Tables 7, 8, where the target coverage is $(1 - \alpha) = 0.8$ as before. For the breast cancer dataset, CB has a very small misclassification rate, but almost 19% of all prediction sets are empty and 0% are both, so the coverage is attained by making either single correct predictions or empty ones. Bayes on the other consists of more misclassifications but fewer uninformative predictions, but the attained coverage is a much higher value of 0.99. For the Parkinson's dataset, CB makes very few uninformative predictions, but has a relatively high misclassification rate. Bayes on the other hand is very conservative, with 31% uninformative predictions, hence the high average length and over-coverage. It is interesting to note the two sorts of behaviours attained by CB, which likely depends on the Bayesian model that was used to construct the CB intervals.

In Figure 3, we see the distributions of $p_i := p(y_i \mid x_i, Z)$ of the Bayesian model with the corresponding CB interval length. We see that for CB intervals of length 1, the values of $p_i$ tend to be heavily skewed towards 0 or 1, which corresponds to the Bayesian model being strongly predictive. For empty CB intervals, in both cases the probability mass is distributed away from 0 and 1; for the breast cancer dataset it is evenly distributed on $(0, 1)$ whereas for Parkinson's it is concentrated around 0.5. When given a CB interval of length 0, it may be more informative to actually return the value $p_i$, which is the corresponding Bayesian prediction.

Table 7: Misclassification rates

| Dataset | Bayes | CB |
|---|---|---|
| Breast Cancer | 0.011 (0.001) | 0.002 (0.000) |
| Parkinson's | 0.064 (0.006) | 0.124 (0.004) |

Table 8: Uninformative Rates

|         |         | Both     |       | Empty        |
| Dataset | Bayes   | CB       | Bayes | CB           |
|---------|---------|----------|-------|--------------|
| Breast Cancer | 0.059 (0.002) | 0.000 (0.000) | 0 | 0.186 (0.005) |
| Parkinson's | 0.312 (0.009) | 0.003 (0.001) | 0 | 0.070 (0.006) |

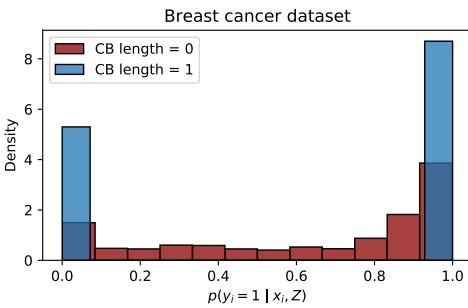 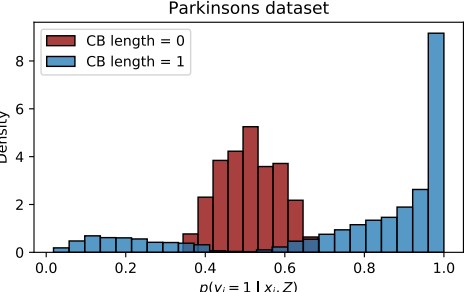

Figure 3: Distribution of $p(y_i \mid x_i, Z)$ on test data for CB intervals of length 1 or 0 for breast cancer (left) and Parkinson's (right).

## D.4 Hierarchical

### D.4.1 Simulated

In this section, we implement the split conformal method on the simulated grouped data example of Section 4.3. We simply carry out split conformal prediction within each group without sharing information between groups, that is we fit the model to $n_j/2$ data points and compute residuals on the remaining $n_j/2$ data points for each group $j$. We use ridge regression with cross-validation as the base model. The results are given in Table 9. As expected, the split method attains close to the target coverage of $0.8$ for both homoscedastic and heteroscedastic scenarios within each group. However, the average lengths are noticeably worse than CB, which is likely due to the small datasets and the lack of sharing information between groups.

Table 9: Simulated grouped dataset for split conformal method Coverage values *not* within 3 standard errors (in brackets) of the target coverage $(1 - \alpha) = 0.8$ are in **red**.

|                |         | Scenario 1 | Scenario 2 |
|                | Group   | Split      | Split      |
|----------------|---------|------------|------------|
| Coverage       | 1       | 0.844 (0.027) | 0.840 (0.027) |
|                | 2       | 0.844 (0.025) | 0.840 (0.024) |
|                | 3       | 0.856 (0.022) | 0.858 (0.022) |
|                | 4       | 0.842 (0.025) | 0.858 (0.025) |
|                | 5       | 0.778 (0.027) | 0.768 (0.027) |
|                | Overall | 0.833 (0.011) | 0.833 (0.010) |
| Length         | 1       | 4.02 (0.27) | 4.84 (0.33) |
|                | 2       | 4.41 (0.35) | 9.20 (0.78) |
|                | 3       | 3.60 (0.16) | 2.81 (0.13) |
|                | 4       | 4.10 (0.20) | 1.33 (0.13) |
|                | 5       | 3.73 (0.22) | 4.06 (0.23) |
|                | Overall | 3.97 (0.11) | 4.45 (0.18) |
| Run-time (secs) | Overall | 0.050 (0.000) | 0.049 (0.000) |

### D.4.2 Radon

Using the same model as in (11), we analyze the radon dataset[1], introduced in Gelman and Hill (2006, Chapter 12). The dataset consists of 919 home radon levels in Minnesota, where the covariate is the location of measurement, with $x = 0$ corresponding to basement and $x = 1$ to the first floor. The groups are the 85 counties in which the homes are located, and vary significantly in group size. Around half of the counties contain $n_j \leq 4$ measurements, with the smallest county containing one value and the largest containing 119.

As many of the group sizes are quite small, we do not repeat train-test splits and evaluate coverage. Instead, we compare the CB and Bayes intervals on the entire dataset for different floor values $x$ and counties, and discuss the effects of $n_j$ on the choice of $\alpha_j$. As each $x \in \{0, 1\}$, we specify $x_{\text{test}}$ as all possible group indicators and predictors, resulting in $85 \times 2 = 170$ test values. For the predictive intervals, we use a grid of size 100 between [-6,6]. MCMC for the radon example required around 156s, and computing the 170 predictive intervals took 0.65s and 2.69s for Bayes and CB respectively, where we have excluded the first run compilation time for JAX.

As we need $\alpha_j \geq 1/(n_j + 1)$ to get intervals that are not the entire real line, we set $\alpha_j = 1.1/(n_j + 1)$ (for numerical reasons) and compare the Bayes and CB intervals. The average CB length is 2.66 compared to 2.17 for the Bayes intervals, noting that we are averaging over all possibilities instead of the distribution of $x_{\text{test}}$. In Figure 4, we plot $\pi_j(y)$ for the two value of $x \in \{0, 1\}$ for two groups. For the group size $n_j = 4$, we see that $\pi_j(y) \geq 0.2$, so any $\alpha < 0.2$ would return us the real line as the confidence set. For $n_j = 52$, the ranks are much smoother, giving us more resolution in the confidence sets with respect to $\alpha$. In Figure 5, we show the rank plots for $n_j = 1$, which only contain the ranks $\{0.5, 1\}$. Interestingly, for county 41, $x = 1$ returns the empty set for $\alpha \geq 0.5$ and the real line for $\alpha < 0.5$, which is a consequence of the small group size. CB is able to return non empty sets for county 49 with $\alpha \geq 0.5$. All CB sets appear to be connected.

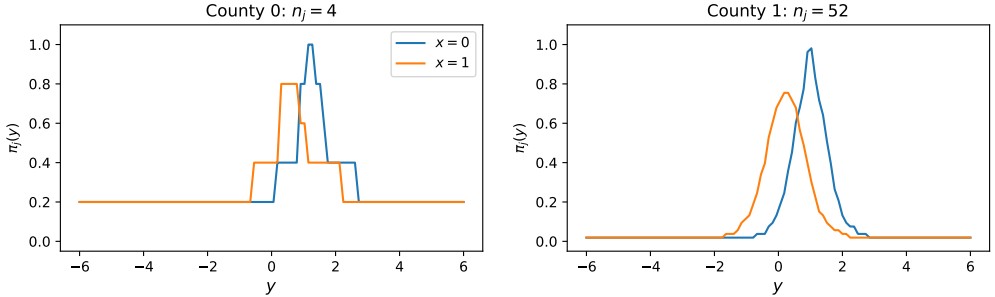

Figure 4: Plot of rank $\pi_j(y)$ for $x \in \{0, 1\}$ with $n_j = 4$ (left) and $n_j = 52$ (right).

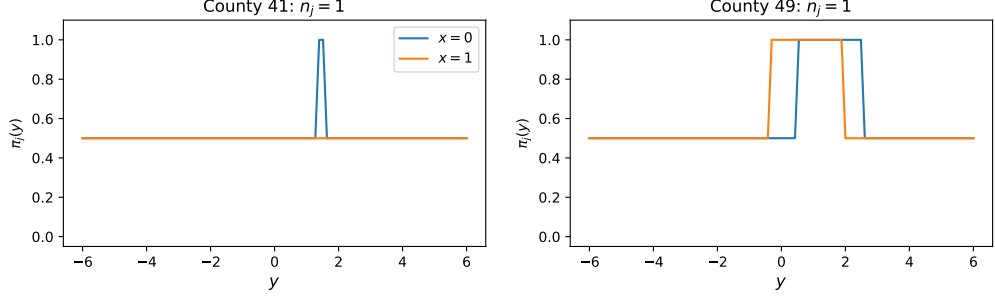

Figure 5: Plot of rank $\pi_j(y)$ for $x \in \{0, 1\}$ with $n_j = 1$ for two groups.

As a reference, fitting a linear mixed-effects model in `statsmodels` (Seabold and Perktold, 2010) to the whole dataset takes around 600ms, so the full conformal method, which would require refitting

---

[1]We base this example on the PyMC3 notebook here: `https://docs.pymc.io/notebooks/multilevel_modeling.html/`

for each of the 100 grid value and 170 test values, would require 170 minutes in total. On the other hand, CB requires much less time and has similar group structure.

## D.5 MCMC Times

In Table 10, we report the average times (and standard errors) for running MCMC on the NC6 virtual machine. We point out that the times for the hierarchical methods are longer as we needed to increase the tuning steps and acceptance probability to prevent divergences in the chains.

Table 10: Run-time in seconds for MCMC

| Dataset | MCMC |
|---|---|
| Diabetes $(c = 1)$ | 21.868 (0.135) |
| Diabetes $(c = 0.02)$ | 26.790 (0.365) |
| Diabetes $(d = 5)$ | 30.825 (0.214) |
| Diabetes $(d = 0.001)$ | 13.166 (0.072) |
| Boston $(c = 1)$ | 22.827 (0.036) |
| Boston $(c = 0.02)$ | 24.362 (0.429) |
| | |
| Breast Cancer | 45.418 (0.804) |
| Parkinson's | 29.239 (0.302) |
| | |
| Scenario 1 | 90.109 (1.605) |
| Scenario 2 | 78.403 (1.544) |
| Radon | 156.087 (0.000) |

## D.6 Grid effects

To quantify the grid effects, we also compute the coverage by directly evaluating $\pi(Y_{n+1})$ for each test point and checking if it satisfies condition (3). Of course in practice this is not possible as we do not observe $Y_{n+1}$.

For the grid conformal method, we compute the $y \in \mathcal{Y}_{\text{grid}}$ that is nearest to $Y_{n+1}$, and report 0 or 1 if this grid value is in the conformal prediction set. Note that this implementation of the grid method can both under and over cover. Denote $\delta$ as the resolution of the grid, and the smallest grid value in the conformal prediction set as $a$. If $a - \delta < Y_{n+1} < a - \delta/2$, we may incorrectly reject $Y_{n+1}$ if it is truly in the set and $a - \delta$ is not. Similarly, if $a - \delta/2 < Y_{n+1} < a$ we can incorrectly accept if $Y_{n+1}$ is not actually in the set but $a$ is. Note that the estimated average length is also affected by this.

We compare the grid and exact method in Tables 11, 12, 13. The largest discrepancy in average coverage is only 0.008, which is quite negligible. However, we expect this discrepancy to increase as $|\mathcal{Y}_{\text{grid}}|$ decreases.

Table 11: Diabetes; Grid versus exact coverage, with target $(1 - \alpha) = 0.8$

| | | CB Grid | CB Exact |
|---|---|---|---|
| Coverage | $c = 1$ | 0.808 (0.006) | 0.810 (0.005) |
| | $c = 0.02$ | 0.809 (0.006) | 0.810 (0.006) |

Table 12: Boston; Grid versus exact coverage, with target $(1 - \alpha) = 0.8$

| | | CB Grid | CB Exact |
|---|---|---|---|
| Coverage | $c = 1$ | 0.800 (0.005) | 0.800 (0.005) |
| | $c = 0.02$ | 0.799 (0.005) | 0.799 (0.005) |

Table 13: Simulated grouped dataset; Grid versus exact coverage, with target $(1 - \alpha) = 0.8$

|  | | Scenario 1 | | Scenario 2 | |
|---|---|---|---|---|---|
|  | Group | CB Grid | CB Exact | CB Grid | CB Exact |
| Coverage | 1 | 0.794 (0.022) | 0.786 (0.023) | 0.786 (0.025) | 0.790 (0.024) |
|  | 2 | 0.812 (0.024) | 0.816 (0.023) | 0.812 (0.024) | 0.818 (0.023) |
|  | 3 | 0.824 (0.022) | 0.820 (0.022) | 0.824 (0.020) | 0.824 (0.020) |
|  | 4 | 0.798 (0.022) | 0.796 (0.021) | 0.836 (0.021) | 0.838 (0.022) |
|  | 5 | 0.810 (0.020) | 0.812 (0.019) | 0.796 (0.022) | 0.792 (0.022) |

# E   Datasets, Licenses and Societal Impact

We demonstrate our examples on 5 datasets, namely the the diabetes dataset (Efron et al., 2004), the Boston housing dataset (Harrison Jr and Rubinfeld, 1978), the Wisconsin Breast cancer dataset (Wolberg and Mangasarian, 1990), the Parkinson's dataset (Little et al., 2008) and the Radon dataset (Gelman and Hill, 2006). The first 3 datasets are available in `sklearn`, the Parkinson's dataset can be found on the UCI machine learning repository[2] (Dua and Graff, 2017), and the Radon dataset is available on Andrew Gelman's website[3]. Details on data acquisition is provided in the relevant references. We verified that the datasets do not contain personally identifiable information or offensive content by manual checking. The package `sklearn` is distributed under the 3-Clause BSD license. JAX and PyMC3 are both distributed under the Apache License, V2.

The conformal method relies on the weak assumption of exchangeability. In terms of negative societal impacts, it may be tempting to apply the method blindly to real world problems without challenging this assumption as it seems quite weak. Applications where calibration is very important but data is not exchangeable would then be at risk.

---

[2] `https://archive.ics.uci.edu/ml/datasets/Parkinson's`
[3] `http://www.stat.columbia.edu/~gelman/arm/examples/radon/`