# OpenReview forum: "Conformal Bayesian Computation"
_NeurIPS.cc/2021/Conference — NeurIPS 2021 Poster_

### Official Review · Reviewer_dTHk · 2021-07-10

**Rating:** 4
**Confidence:** 4

**Summary:**

 The authors develop a fast conformal Bayesian computation approach.  The original conformal prediction method can be computationally intensive:  It often requires refitting the model for every augmented value y. The authors avoid this refitting step by using importance sampling. The authors also show the extension to conditionally exchangeable data, which is especially useful for hierarchical data. The resulted conformal interval has finite sample correct coverage and is immune to model misspecification or distribution assumptions.

**Limitations And Societal Impact:**

The  authors have adequately addressed potential negative societal impact of their work.
I have listed more limitations in my main review.

**Main Review:**

The paper is well structured and easy to follow. I also agree with the authors that prediction calibration is crucial in the M-open world, which is arguable always the case. The prospered conformal Bayesian computation method sheds some light on this direction.

However, I am concerned about whether the novelty of the proposed method can be distinguishable from its existing building blocks. Conformal inference/distribution-free Predictive Inference—and thereby the main algorithm considered in this paper—are not new ideas. It is also not a new idea to use (regularized) importance sampling to avoid model refitting and obtain posterior integrals in the context of either the leave-data-out (LOO) or the add-one-data-in (AOI, e.g. Bürkner et al. 2020 used this importance-sampling-AOI idea in time series).  It appears the only methodology contribution this paper is making is to apply the importance sampling strategy to the model refitting step in conformal Bayesian inference. I am not convinced that such incremental contribution would grant a publication.

In terms of its practical usefulness, I have two concerns. First, throughout the paper, the authors seem to emphasize the situation in which x_{n+1} is fixed. But this is not a typical perdition task in which we have a known x_{n+1}. In the words of Lei  (2018): “In some applications, where Xn+1 is not necessarily observed, prediction intervals are build by evaluating 1{y ∈ Cconf (x)} over all pairs of (x,y) on a fine grid“. Perhaps x_{n+1} can also be evaluated on a grid, but then that method is rather related to jackknife-conformal-inference, which itself can be computed by importance sampling.

Second, the “correct frequentist coverage” is a probabilistic statement averaged over all observed data and x_{n+1}. This does not mean the interval estimate derived from conformal Bayes is exact for y_{n+1} conditional on a fixed x_{n+1}, nor necessarily optimal for interval predictions (optimal in the sense of interval scoring rules). Think about the Jackknife case in which the bandwidth is a constant across x_i. I suggest the author clearly state the definition of frequentist coverage as not all audience in the machine learning community can immediately tell the subtle difference.

Minor:
* L-82: why is the rank normalized by n+1? I think the rank statistic should the integer.

* How is the grid of y chosen? Could you use Bayes prediction to design the grid?

* What is the advantage of AOI against jackknife used in conformal Bayes? Is the only difference that IS-AOI is computationally easier than IS-LOO?

* “The CB and Bayes methods have comparable run-times” I think it is misleading. CB can have an arbitrarily larger running time if x_{n+1} is evaluated at many values.

* Overall, the experiment section is relatively weak. The sample sizes in all three examples are too small to reflect a modern big data challenge. It is not common to see all MCMC running time < 1 second from ML conferences.

* L-336: I think “leverage” is the wrong concept here. The leverage only depends on x. Here you have the influence of y too.

* "If only approximate posterior samples are available, e.g. through variational Bayes (VB), then an AOI scheme may still be feasible, where one includes an additional correction term in the IS weights for the VB approximation". I disagree. Note that Magnusson et al. (2019) are on the LOO of exact Bayes when there are only VB samples. In your context, you need not only adjust for the VB approximation for sigma_{n+1}, but also all sigma_{1}, ... sigma_{n} for lack of exact posteriors. Such a task is generally not feasible.


**Time Spent Reviewing:**

7

---

> ### Author Response · Authors · 2021-08-10
> **Response to Reviewer dTHk**
>
> We would like to thank the reviewer for their feedback on our submission. We agree that our method is simple, yet powerful, and we see this as a strength, particularly for practical adoption. The approach is widely applicable and uses a novel combination of Bayes, importance sampling and full conformal inference. This may seem obvious in hindsight, but has not been explored in the literature to the best of our knowledge.
>
> We can summarise our main contributions as follows. For the $\mathcal{M}$-open Bayesian, CB is a general wrapper around standard MCMC model fitting that provides inexpensive post-calibration of predictive intervals with finite sample guarantees of validity. For the frequentist, CB demonstrates that full conformal inference is feasible for a large class of models, which was not previously possible. Finally, we provide extentions to conformal inference for hierarchical Bayesian models, which are widely used for information sharing between groups.
>
> We provide a point-by-point response below.
>
> 1. It is important to note that $X_{n+1}$ is random, not fixed, as we state in Equation (2). This is vital for the conformal prediction guarantees to hold, as the frequentist probability statements are over the joint distribution of $Z_{1:n+1}$, where $Z_i = (Y_i,X_i)$.
>
>     Nonetheless, in the cases where $x_{n+1}$ is not known beforehand, or the analyst wishes to compare their Bayesian and frequentist predictions, the intervals can indeed be evaluated on the $x$-grid (implemented in parallel if desired). It is simply a matter of computing a CB interval for each $x_{n+1}$, so computation is linear wrt the size of the $x$-grid. In fact, in our experiments, the test sets are of size $0.3 n$ and thus consist of multiple test points. We observe that computation times are still very fast for multiple $x$ values. This can be additionally sped-up by running in parallel across multiple GPUs.
>
>     Unfortunately, we do not follow the connection between evaluating $x_{n+1}$ on a grid and LOO conformal inference, nor do we see the difficulties in computing a separate interval for each $x_{n+1}$.
>
> 2. We provide a definition of the CB frequentist coverage in Equation (2) in the main paper, emphasizing that the distribution is over $Z_{1:n+1}$  and hence not conditional on the realisation $x_{n+1}$. We will stress this again in the discussion.
>
> ### Minor
> 1. The rank is normalized by $n+1$ so that it is on the same scale as $\alpha$. Our definition of the conformal interval follows equation (2.18) in Vovk et al. (2005).
>
> 2. The grid is of size 100 evenly spaced between $[y_{\text{min}}-2, y_{\text{max}}+2]$, where the minimum and maximum is from the dataset $y_{1:n}$ as discussed in Section 4. We have considered using the Bayesian predictive to design the grid, but it may be difficult under model misspecification. More details on grid selection can be found in Chen et al. (2016, 2018), but we have opted for the simpler choice here. An evaluation of the accuracy of the grid method is provided in our Appendix D.6. Thank you for raising this, we will provide further discussion for choosing $\mathcal{Y}_{\text{grid}}$ in the main body of the paper.
>
> 3. The main advantage of AOI versus LOO is that IS weights are more stable for AOI than LOO, as AOI is predicting *in-sample* and the target distribution has thinner tails than the proposal. Computation for AOI versus LOO is also simpler and faster, as we require smaller intermediate arrays for the IS weights. We have checked the above empirically and will now include an empirical comparison of the AOI/LOO IS weights in the Appendix.
>
>     LOO may be useful if there are significant outliers, as discussed in Page 28, Vovk et al. (2005), and it is simple to extend CB to work with LOO using IS. Otherwise, the intervals from LOO and AOI will be similar; we have checked this empirically and will include this in the Appendix. Including all $n+1$ data-points in the fitted model is also quite standard for convenience, e.g. Algorithm 1 of Lei et al. (2018) and second-last paragraph on Page 10 of Zeni et al. (2020). Thank you for raising this, we will provide this discussion in the paper.
>
> 4. Conventional Bayes using MCMC outputs also has arbitrary run times (linear in the size of the $x$-grid) as the posterior predictive distribution function, and its intervals, need to be estimated via Monte Carlo for each $x_{n+1}$. It is not obvious to us how this is a misleading comparison.
>
> 5. We believe the sample sizes are relevant to many real-world Bayesian settings, where the prior plays a role. The MCMC overhead running times are not $<1$s. This is stated in the paper in Section 4, e.g. $\approx 80$-$90$s for the hierarchical example in Lines 306-307. The tabular run-time values are for computing the intervals *after* MCMC, noting that the MCMC already has to be run for standard Bayes. We will clarify this further in the Tables.
>
> 6. Thank you, we agree that the use of 'leverage' was a poor choice of words. We simply mean that $(y,X_{n+1})$ has a large effect on the posterior. We will update the wording.
>
> 7. We respectively disagree. In our AOI method, we are already correcting  $\sigma_1,\ldots,\sigma_n$ in addition to $\sigma_{n+1}$ to refit the posterior to include $(y,X_{n+1})$, which we observe is feasible. In the VB approximation case, we would need to include an additional factor proportional to the true posterior density divided by the VB posterior density to each importance weight; we do not see how this significantly affects feasibility beyond IS weight stability and intermediate arrays for the weights. It is true however that a more thorough examination of the stability of IS weights would be needed, and we will comment on this in the paper.
>
>
> ### References
>
> Vovk, V., Gammerman, A., \& Shafer, G. (2005). Algorithmic learning in a random world. Springer Science \& Business Media.
>
> Chen, W., Wang, Z., Ha, W., \& Barber, R. F. (2016). Trimmed conformal prediction for high-dimensional models. arXiv preprint arXiv:1611.09933.
>
> Chen, W., Chun, K. J., \& Barber, R. F. (2018). Discretized conformal prediction for efficient distribution‐free inference. Stat, 7(1), e173.
>
> Lei, J., G’Sell, M., Rinaldo, A., Tibshirani, R. J., \& Wasserman, L. (2018). Distribution-free predictive inference for regression. Journal of the American Statistical Association, 113(523), 1094-1111.
>
> Zeni, G., Fontana, M., \& Vantini, S. (2020). Conformal prediction: a unified review of theory and new challenges. arXiv preprint arXiv:2005.07972.

---

> > ### Comment · Reviewer_dTHk · 2021-08-19
> > **thanks for clarification**
> >
> >
> > I thank the authors for clarification.  I have a minor reply to
> >
> > > we do not see how this significantly affects feasibility beyond IS weight stability and intermediate arrays for the weights.
> >
> > If such IS step is feasible, we will always obtain exact posterior integrals from VB.

---

> > > ### Author Response · Authors · 2021-08-19
> > > **Response to Reviewer dTHk**
> > >
> > > Thank you for the reply. That is a good point - the IS weight stability is strongly dependent on the quality of the VB approximation, which may be poor. We may need a larger number of IS samples $T$, or significant effort in stabilizing the problematic IS weights as in Magnusson et al. (2019). We note that Magnusson et al. (2019) explicitly consider IS for LOO with a VB proposal and show that it can work.
> > >
> > > We will add a strengthened statement to this effect in the discussion, making clear the additional computation also required from Pareto-smoothing, and acknowledge your comment from an anonymous reviewer.

---

### Official Review · Reviewer_ijh8 · 2021-07-13

**Rating:** 7
**Confidence:** 4

**Summary:**

The authors present an approach to using a Bayesian model to provide predictive intervals with  frequentist coverage within the framework of conformal prediction.  This work generalizes existing Bayesian methods for conformal inference that apply only to conjugate models.  The authors provide a straightforward generalization of their approach to partially exchangeable data.  Empirical validation on several datasets, and comparison to a baseline approach is provided.

**Limitations And Societal Impact:**

The authors adequately addressed the limitations and potential negative societal impact of their work.

**Main Review:**

The approach to extending conformal intervals beyond conjugate models is a sensible, simple and novel contribution.  The overall exposition and description of the method are very clear.   The work appears potentially significant and may be impactful within the Bayesian community by virtue of its simplicity and clear exposition.

However the benefits of the conformal Bayes in comparison to the broader space of non-Bayesian methods for conformal inference and methods for constructing prediction intervals beyond conformal prediction are not clear; it is not clear what (if anything) the authors provide which is not provided by Barber et al. (2021), for example.  A more detailed comparison to existing conformal prediction methods (both in pros and in experimental validation) is needed.

Additionally, the use of the “add one in” (AOI) posterior predictive density as the conformity score was surprising to me.  Can the authors clarify when this will and will not work usefully as a conformity score?

Assuming I understand the set-up correctly, the score will break down, for example, when predicting with a noiseless Gaussian process regression:
theta ~ GP(0, k)
X_i   \overset{i.i.d.}{\sim} N(0,1)
Y_i = theta(X_i)
In this case, the density in the first display equation of section 2.2 does not exist (i.e. you cannot define it w.r.t Lebesgue) because for any i satisfying 1<i <= n+1, theta(X_i) | X_i,  Z_{1:n+1} is no longer random once we condition on Z_{1:n+1}.

In any case, the authors would strengthen the paper by clarifying what is meant by the first display equation of section 2.2; if since (X_i, Y_i) is an element of Z_{1:N+1} (following lines 75-76 of the paper), how should p(Y_i | X_i, Z_{1:N+1}) be interpreted?



Relatedly, the preference for AOI predictive densities rather than leave one out densities was surprising to me.  And the discussion of this choice on lines 123-131 appeared to me as a bit of a red herring; the three bulleted claims are neither obvious nor supported with evidence or references.  Could the authors comment on whether these observations are part of their contribution, or if these differences to LOO are already known?  In any case, I would have appreciated an empirical comparison to LOO predictive densities used to define the conformity score (again using IS).

It would greatly strengthen the paper to include (as a theorem) explicit conditions under which the CB intervals have the correct coverage.  Such a statement would help the practitioner/reader understand when the method should be applied.


**Time Spent Reviewing:**

2.5 hours

---

> ### Author Response · Authors · 2021-08-10
> **Response to Reviewer ijh8**
>
> We would like to thank the reviewer for their feedback on our submission. We provide a point-by-point response below.
>
> 1. While our method is primarily aimed at providing a wrapper for conventional Bayesian analysis (using standard MCMC output), we believe CB provides a computationally feasible method to compute full conformal intervals for a larger class of models (compared to the existing  restrictive examples of full conformal before, such as ridge/Lasso). We believe the comparison of full conformal inference versus other conformal methods is slightly beyond the scope of this paper. We expect other methods such as the jackknife+ to lie in between the full and split method in terms of the length/computation trade-off. We will provide further comparisons and a discussion in the Appendix.
>
> 2. We will add the explicit definition of the AOI density, that is
>     $
>     p(y \mid x, Z_{1:n+1}) = \int f_\theta(y \mid x) \pi(\theta \mid Z_{1:n+1})  d\theta.
>     $
> To clarify, the predictive density $p(y \mid x, Z_{1:n+1})$ is defined as above, where we plug in $y = Y_i$ and $x = X_i$ for $\sigma_i$ as in Line 101.
>
>     Thank you for the very interesting noiseless GP example, this is a rare case where LOO works and AOI does not. We will discuss this in the Appendix (crediting an anonymous referee). Please see the next point for AOI vs LOO in general.
>
>
> 3. The choice of AOI is similar to including $(Y_i,X_i)$ for $\sigma_i$ when fitting the predictive model $\hat{\mu}$ in regular conformal inference. It is often convenient and standard to include $(Y_i,X_i)$ in the fitted model for residual $\sigma_i$ in full conformal inference, e.g. Algorithm 1 in Lei et al. (2018) and the second-last paragraph on Page 10 of Zeni et al. (2020).  The motivation for a LOO conformity score is given in Page 28, Vovk et al. (2005), where deleting $(Y_i,X_i)$ may be helpful in the presence of extreme outliers.
>
>     In general, the intervals from LOO and AOI will be quite similar, and CB can easily be extended to work with LOO using IS. However, the AOI IS weights will be more stable, as our posterior is fit to include $(Y_i,X_i)$ so we are predicting *in-sample*. The posterior with $n+1$ points also generally has thinner tails than with $n$ points, which is desirable for IS weight stability. Computation for AOI versus LOO is also simpler and faster, as we require smaller intermediate arrays for the IS weights. We have verified the above empirically and will provide a comparison of the AOI/LOO IS weights and intervals in the Appendix as suggested.
>
>     Unfortunately, as spotted by another reviewer, there is a typo in Line 123 - it should be 'thicker' instead of 'thinner'. This discussion in lines 123-131 is 'new' in the sense that the Bayesian is often not predicting in-sample, and so its properties are not usually discussed. However, it is a natural analogy to the discussion provided in Section 2.1.1 of Vehtari et al. (2017). Thank you for raising this, we will provide this clarification in later versions of the paper.
>
> 4. One of the remarkable features of conformal inference is that it has correct coverage so long as exchangeability holds. This applies to CB too, which is a special case. The theorem is due to Vovk et al. (2005), which we will refer to explicitly.
>
>
>
> ### References
>
> Lei, J., G’Sell, M., Rinaldo, A., Tibshirani, R. J., \& Wasserman, L. (2018). Distribution-free predictive inference for regression. Journal of the American Statistical Association, 113(523), 1094-1111.
>
> Zeni, G., Fontana, M., \& Vantini, S. (2020). Conformal prediction: a unified review of theory and new challenges. arXiv preprint arXiv:2005.07972.
>
> Vovk, V., Gammerman, A., \& Shafer, G. (2005). Algorithmic learning in a random world. Springer Science \& Business Media.
>
> Vehtari, A., Gelman, A., \& Gabry, J. (2017). Practical Bayesian model evaluation using leave-one-out cross-validation and WAIC. Statistics and computing, 27(5), 1413-1432.

---

> > ### Comment · Reviewer_ijh8 · 2021-08-15
> > **Reply to Rebuttal**
> >
> > Thank you for your thoughtful and detailed reply.  Replying point by point below.
> >
> > 1. I am glad to hear you will be adding a discussion in the appendix.  It will be nice to read, for example, how one using a Bayesian workflow might choose between conformal intervals and posterior predictive intervals.
> >
> > 2. I am glad to hear an explicit density will be added.  I think this will strengthen the paper.  It would be valuable as well to understand what makes this “rare case” with the GP different.  If I were to use this method, how might I know I am not in a similar “rare case”.  This is why I think explicit theory would be useful in the present paper — even if it is simply adapted from / taken from another source (with attribution).
> >
> > 3.  Great!
> >
> > 4. If I were using this method, I would appreciate a reproduction of this theorem in the paper (or appendix) so that I wouldn’t need to look it up / catch up with any differences in notation.  See my reply to 2.
> >
> > I will raise my score by 1 point.

---

> > > ### Author Response · Authors · 2021-08-16
> > > **Response to Reviewer ijh8**
> > >
> > > Thank you for the helpful suggestions. We agree with the points raised and will include them in a final version.

---

### Official Review · Reviewer_mkBe · 2021-07-15

**Rating:** 6
**Confidence:** 2

**Summary:**

This paper proposes a scalable method for conformal Bayesian prediction, by estimating the modified posterior predictive density using "add-one-in" importance sampling.  On several tabular datasets with sparse and hierarchical models, estimation is shown to be efficient for non-extreme miscoverage levels, and the produced credible intervals have good coverage.

**Post-rebuttal update:** Thank you for the response.  I am keeping my score unchanged: on the plus side, the method is sensible and easy to implement; on the other hand, there are some (understandable) limitations when scaling to large datasets or applied on approximate posterior samples; and some reviewers raised questions about novelty, which I'm not most suited to judge, as I'm not an expert in this field.  In aggregate, the rating of "marginally above the acceptance threshold" still describes my opinions well.

**Limitations And Societal Impact:**

The methodological limitations (requiring the original posterior to be a good IS proposal) is discussed in the paper.

As mentioned above, the work will be strengthened if there are more empirical comparisons of scalable conformal prediction methods using alternative scores.

Potential societal impacts are adequately discussed.

**Main Review:**


The proposed method assumes that a large number of samples from the (original) posterior can be produced efficiently, after which prediction on a single test input can be carried out in O(N_{samples} * N_{train} * N_{grid}) time.  While there is no guarantee on the number of posterior samples needed, diagnostics is possible by looking at the effective sample size.  As the method is easy to implement and diagnostics is always possible, it can be a sensible choice as a "first attempt" at moderately-sized problems, especially when Bayesian modeling is also being considered.

Still, the time complexity above can still be expensive for large-scale datasets.  Also the efficiency of the importance sampling procedure is not fully clear, even though the authors provided some informal discussion: as reported in the first experiment, importance sampling is more difficult when the desired miscoverage level is more extreme, or when the prior is misspecified; the latter setting can be relevant in practical applications.  It is also unclear how the sample size requirement changes as the model becomes more complex.

The experiments demonstrate consistent improvement over standard Bayesian modeling, but it is not fully clear if the method outperforms other scalable conformal prediction methods based on alternative scores: for example, in the first two experiments, the split conformal method has comparable or only slightly worse (CI 5% wider) performance than the proposed method, while being significantly cheaper computationally.  It will strengthen the paper if the authors provide more comparison between these two methods, by e.g., evaluating the split method on a grouped dataset.

The paper is well written, I didn't find any unsubstantiated claims, and it seems that relevant prior work has been discussed; although I'm not an expert in this field, so I'm not fully confident.

Minor: L123: thinner -> thicker



**Time Spent Reviewing:**

4

---

> ### Author Response · Authors · 2021-08-10
> **Response to Reviewer mkBe**
>
> We would like to thank the reviewer for their feedback on our submission. We provide a point-by-point response below.
>
> 1. Importance sampling can scale poorly as the dimension of the parameter space increases, but there are methods to mitigate this effect, such as the pareto-smoothing approach of Vehtari et al. (2015).   We will include additional guidance on choosing $T$ in the main body of the paper.
>
> 2. We believe a thorough discussion of the merits of full conformal inference in comparison to other conformal methods is slightly beyond the scope of this paper, as we are focused on CB being a wrapper around conventional Bayesian analysis (using standard MCMC output). However, we highlight that given MCMC samples, the full conformal method is actually more practical than the split method for CB. We will include a full comparison to the non-Bayesian split-method for the grouped dataset and additional comparisons in the Appendix as suggested.
>
> 3. Thank you for spotting the typo, we will correct this.
>
> ### References
>
> Vehtari, A., Simpson, D., Gelman, A., Yao, Y., \& Gabry, J. (2015). Pareto smoothed importance sampling. arXiv preprint arXiv:1507.02646.

---

### Official Review · Reviewer_tEpY · 2021-07-16

**Rating:** 4
**Confidence:** 2

**Summary:**

This paper introduces Bayesian and importance sampling techniques into a previously known conformal prediction framework.

I had troubles identifying what exactly is the novel contribution of this paper. It could use a brief section stating "our own contributions are A, B , C". In my understanding the conformal prediction framework was known before and the novel items are using the Bayesian posterior predictive density as a conformity score and additionally estimating the predictive distribution given new data from a weighted importance sampling where the weights are just the likelihoods of the prameter draws based on the new data point.

There are modelling extensions from completely exchangeable models to partial exchangeability, i.e., where we have groups and complete exchangeability is still valid in the groups, but not outside the groups. From this flat hierarchical structure there is a further extension to hierarchical models that may model deeper hierarchies and via hyperpriors may share information between groups.



**Main Review:**

I think the paper describes fairly straightforward ideas and there is not much surprising or interesting work here. I might be misunderstanding or missing some points, but the main questions that arise are not even attempted to be answered.

For instance the discretization into a discrete finite grid: What is its size, or more specifically, what is the trade-off between its size and the accuracy of approximation. The same question arises for me when I think of the parameter T for approximating an integral by an empirical average. how large should I choose T such that the error is within an eps error or within a multiplicative (1+-eps) error? Shouldn't the sample size be highly dependent on the dimension d of the parameters? I mean like a large polynomial in d or even 2^d.

The answer given by the paper is that those values are constants so it's easy for me to say that the paper introduces arbitrarily complicated models and their algorithm is only a heuristic without any guarantees. I think the modeling efforts are nice but too straightforward for a top conference like this. The only theoretical claim/proof is also more like an observation.

In the experimental part the distributions are concretized with quite simple distributions. Would it be possible to quantify the computational efforts here in those special cases?

**Time Spent Reviewing:**

6

---

> ### Author Response · Authors · 2021-08-10
> **Response to Reviewer tEpY**
>
> We would like to thank the reviewer for their feedback on our submission. We agree that our method is simple, yet powerful, and we see this as a strength, particularly for practical adoption. The approach is widely applicable and uses a novel combination of Bayes, importance sampling and full conformal inference. This may seem obvious in hindsight, but has not been explored in the literature to the best of our knowledge.
>
> We can summarise our main contributions as follows. For the $\mathcal{M}$-open Bayesian, CB is a general wrapper around standard MCMC model fitting that provides inexpensive post-calibration of predictive intervals with finite sample guarantees of validity. For the frequentist, CB demonstrates that full conformal inference is feasible for a large class of models, which was not previously possible. Finally, we provide extentions to conformal inference for hierarchical Bayesian models, which are widely used for information sharing between groups.
>
> We provide a point-by-point response below.
>
> 1. In our experiments, we have used a grid size of $100$, as discussed in Section 4, e.g. Lines 243-244. We demonstrate in Appendix D.6 that our grid is sufficiently accurate in our experiments in terms of providing the same coverage as an exact method. Selecting the grid is a general problem in full conformal inference, and not just specific to CB. We have opted for a simpler choice here to focus on using the AOI predictive, but more details on how to select the grid for full conformal inference can be found in Chen et al. (2016, 2018). Thank you for raising this. We will add more guidance on choosing $\mathcal{Y}_{\text{grid}}$ in the main body of the paper.
>
>
> 2. The number $T$ of MCMC samples for IS needed to accurately compute integrals is a general issue in Bayesian statistics without a definitive answer, but there exists standard guidance in the literature for addressing this question. In brief, it is both dependent on the mixing (and resulting effective sample size, $\text{ESS}_{\text{MCMC}}$) and the loss in sample size due to IS. We suggest computing the ESS as in Figure 1, which gives Monte Carlo error $\propto 1/\sqrt{\text{ESS}}$. We are also free to estimate the IS variance using standard methods. Please see Chapter 9.3, Owen (2013) for further discussion of standard methods. In general, IS can sometimes scale poorly with dimensionality, but there are also methods to combat this such as Pareto-smoothing (Vehtari et al. (2015)). Thank you for raising this. We will include additional guidance on choosing $T$ in the main body of the paper.
>
> 3. The models chosen in our experiments are relatively standard for Bayesian inference and, as noted above, our method provides a wrapper around standard Bayesian posterior sampling. There is a single computational overhead due to MCMC, but in practice this will already have been carried out for the conventional Bayesian analysis. Our examples are all non-conjugate which we believe to be a fair demonstration, as it generalizes to other Bayesian models. Given the MCMC samples, the bulk of the computation for the CB intervals is the large matrix multiplication of IS weights, which is not dependent on model complexity beyond the cheap likelihood evaluations (for a fixed $T$).
>
> ### References
> Chen, W., Wang, Z., Ha, W., \& Barber, R. F. (2016). Trimmed conformal prediction for high-dimensional models. arXiv preprint arXiv:1611.09933.
>
> Chen, W., Chun, K. J., \& Barber, R. F. (2018). Discretized conformal prediction for efficient distribution‐free inference. Stat, 7(1), e173.
>
> Owen, A. B. (2013). Monte Carlo theory, methods and examples.
>
> Vehtari, A., Simpson, D., Gelman, A., Yao, Y., \& Gabry, J. (2015). Pareto smoothed importance sampling. arXiv preprint arXiv:1507.02646.

---

> > ### Comment · Reviewer_tEpY · 2021-09-01
> > **Thanks for some clarifications**
> >
> > Dear authors, I will leave my score unchanged but also my confidence score remains low since I am no expert in this field.
> >
> > Briefly going over the submission again, I see incremental modifications of existing approaches, no actual analysis of the approximations obtained and no quantification of the trade-off between accuracy and efficiency (computational complexity) wrt the main parameters.
> >
> > The response tends to outsource the problems to other papers, which in my opinion would not raise the strength of the submission. E.g. added discussions or guidelines on the choice of T and Y_grid would be the very least requirement. To strengthen the contribution they should be analyzed with mathematical rigor instead (at least in some nice and some not so nice special cases).
> >
> > Overall I think the paper is well-written and contains nice ideas, but there is not enough substantial contribution for a top conference.

---

### Official Review · Reviewer_Ph2T · 2021-08-03

**Rating:** 7
**Confidence:** 4

**Summary:**

This article proposes a novel approach to constructing prediction intervals with approximately correct frequentist coverage, even on misspecified models.  The basic idea is to employ the conformal prediction technique using the Bayesian posterior predictive density as the conformity (goodness-of-fit) measure.  The main innovation of the article is to use an importance sampling approximation to the posterior predictive density at candidate values of outcome $y$.  This leads to a computationally fast and generally applicable algorithm for using samples from the standard Bayesian posterior to construct well-calibrated prediction intervals, even under misspecification of the Bayesian model.  An extension to hierachical models is also provided, and empirical results are presented demonstrating the performance of the method.

**Ethical Concerns:**

No ethical concerns.

**Limitations And Societal Impact:**

Yes.

**Main Review:**

This is an elegant idea.  The innovation of using importance sampling is pretty simple, but it works out really nicely and I find it quite compelling.  The paper is clear and well-written.  I especially appreciate the clear and concise explanation of the full conformal prediction technique in Section 2.1.  In terms of significance, I could see this becoming a standard part of predictive analysis when using Bayesian models -- it is both powerful and seems easy to implement.

COMMENTS
1. How should one choose $T$ (the number of importance samples)?  How should one choose $\mathcal{Y}_\mathrm{grid}$ (the grid of $y$ points)?  The choices of $T$ and $\mathcal{Y}_\mathrm{grid}$ used in the examples are made without much explanation.  More guidance would be helpful here.
2. On lines 333-337, the paper mentions that conformal Bayes is limited by Monte Carlo error and the quality of the posterior predictive as a proposal distribution.  Does the severity of misspecification affect coverage?  Are there other factors that affect the coverage performance?
3. What role does the choice of model have on the quality of the prediction intervals obtained using conformal Bayes?
4. Does conformal Bayes provide a mechanism for diagnosing misspecification?




**Time Spent Reviewing:**

3

---

> ### Author Response · Authors · 2021-08-10
> **Response to Reviewer Ph2T**
>
> We would like to thank the reviewer for their feedback on our submission. We provide a point-by-point response below.
>
> 1. There isn't a definitive answer for specifying the number $T$ of MCMC samples used for IS, but there is standard guidance in the literature for assessing this. It is both dependent on the mixing (and resulting effective sample size,  $\text{ESS}_{\text{MCMC}}$) and the loss in sample size due to IS. We recommend evaluating the ESS as in Figure 1 to get the desired precision in the density estimates (in the sense of Monte Carlo error $\propto 1/\sqrt{\text{ESS}}$), or estimating the IS variance directly. More details can be found in Chapter 9.3, Owen (2013) and Vehtari et al. (2015).
>
>     For the size of the grid, we recommend trying a few different values and checking if the CB intervals are noticeably different, especially at the edges of the interval. We demonstrate in Appendix D.6 that $|\mathcal{Y}_{\text{grid}}| = 100$ is sufficiently accurate in our experiments. More details on how to select the grid for full conformal inference can be found in Chen et al. (2016, 2018), but we have opted for the simpler choice here.
>
>     Thank you for raising this. We will add more discussion for choosing $T$ and $\mathcal{Y}_{\text{grid}}$ in the main body of the paper.
>
> 2. Model misspecification would not affect the coverage properties as the conformal framework protects from this, but the severity of misspecification would affect the stability of estimates of the AOI predictive density due to Monte Carlo error. That said, the CB intervals are computed from the *ranks* and not the exact values so we expect some robustness to Monte Carlo error. We will add a comment to this effect.
>
>     If the observations are not truly exchangeable, then the coverage guarantees no longer hold. However, this is not specific to CB, as all conformal methods require this assumption. We will re-emphasise this in the discussion.
>
> 3. Intuitively, a poor predictive model will increase the values of the nonconformity scores (the residuals) and hence we'd expect the confidence intervals to be wider, e.g. remark 2 of Lei et al. (2018) for the non-Bayes case.
>
> 4. If the CB intervals are significantly different to the posterior predictive intervals, or the importance weights are extremely unstable, then this likely points to model misspecification, as well-specified Bayesian models are generally well-calibrated. However, we point out that it is difficult to compare Bayesian and frequentist intervals for a single dataset.
>
> ### References
>
> Owen, A. B. (2013). Monte Carlo theory, methods and examples.
>
> Vehtari, A., Simpson, D., Gelman, A., Yao, Y., \& Gabry, J. (2015). Pareto smoothed importance sampling. arXiv preprint arXiv:1507.02646.
>
> Chen, W., Wang, Z., Ha, W., \& Barber, R. F. (2016). Trimmed conformal prediction for high-dimensional models. arXiv preprint arXiv:1611.09933.
>
> Chen, W., Chun, K. J., \& Barber, R. F. (2018). Discretized conformal prediction for efficient distribution‐free inference. Stat, 7(1), e173.
>
> Lei, J., G’Sell, M., Rinaldo, A., Tibshirani, R. J., \& Wasserman, L. (2018). Distribution-free predictive inference for regression. Journal of the American Statistical Association, 113(523), 1094-1111.

---

### Decision · Program_Chairs · 2021-09-27

**Decision:**

Accept (Poster)

**Comment:**

This article presents a novel method for constructing prediction intervals with correct frequentist coverage, based on combining frequentist ideas from conformal prediction with a Bayesian model.  The main idea is to use the Bayesian posterior predictive density as the conformity measure in the standard conformal prediction framework, and employ importance sampling to avoid having to re-fit the model for every data point.  The method uses "add-one-in" importance sampling rather than leave-one-out, which is advantageous in that it lowers the variance of the importance weights, leading to more stable results.  The computational cost is also significantly reduced by approximating the outcome space with a discrete grid of points, following Chen *et al* (2018).

The idea is relatively simple but very generally applicable, and has the potential to be high impact in the Bayesian community.  The simplicity with which it can be implemented is attractive, and the computational cost is modest: it takes less than 2x the time as the standard Bayesian technique (Tables 1-3). Further, the performance guarantees are quite general, and the experiments demonstrate a striking improvement in coverage performance compared to standard Bayes (Tables 1-3). The paper is well-written and sound.

In my view, the main limitations are:

1) Novelty. The method combines standard techniques for conformal prediction, importance sampling, and Bayesian inference.  Thus, the constituent parts are not novel, and they are integrated in a fairly straightforward way.  The novelty lies in bringing these pieces together and recognizing its utility.

2) Need to appropriately choose the grid of $y$ points, $\mathcal{Y}_\mathrm{grid}$.  The paper provides a reasonable default recommendation for this, which is to use 100 evenly spaced points between $y_\mathrm{min}-2$ and $y_\mathrm{max}+2$.  However, it seems that this choice may require application-specific tuning.

3) Model misspecification may increase the variance of the importance sampling approximation, resulting in more unstable inferences.

There was a fairly wide range in the reviewers scores.  There were two ratings of 4 from reviewers tEpY (with a low confidence of 2) and dTHk (with a medium confidence of 4).  I have carefully examined the paper, the reviews, and the authors' reply, and I believe the main criticisms can be addressed as follows:

- Regarding novelty, reviewers tEpY and dTHk criticize the article for providing incremental improvement upon existing approaches.  While it is true that the method can be viewed as combining existing methods, I would argue that the utility of the proposed method —  in comparison with existing alternatives — justifies its simplicity.  Reviewer dTHk also states that Bürkner *et al* (2020) already introduced the idea of using add-one-in importance sampling for approximating posterior integrals; however, I have examined the Bürkner *et al* (2020) paper and while they use importance sampling, I do not see any use of a technique resembling the method in the present paper.

- Regarding the choice of $\mathcal{Y}_\mathrm{grid}$:  Reviewer tEpY criticizes the article for outsourcing the problem of choosing the grid, and the associated tradeoff between computational cost and accuracy, to Chen *et al* (2018).  Since the use of a discrete grid for conformal prediction is not the main novel contribution of the paper, this outsourcing seems reasonable in my opinion.

- Reviewer dTHk is concerned that the assumption of a known $x_{n+1}$ may not be practical, and discusses making predictions for a grid of $x_{n+1}$ points.  However, as the authors state in their reply, the coverage guarantees are for random $X_{n+1}$, not a given fixed $x_{n+1}$. Thus, in fact, the method effectively considers all possible $x_{n+1}$ values.

- Additionally, reviewer dTHk felt that the article should emphasize more clearly that the coverage guarantees hold in expectation, when averaging over the entire dataset as well as the future datapoint $x_{n+1}$.  This is a good suggestion, which is easily addressed via a minor edit, which the authors have volunteered to do in their reply.


Bürkner, P. C., Gabry, J., & Vehtari, A. (2020). *Approximate leave-future-out cross-validation for Bayesian time series models.* Journal of Statistical Computation and Simulation, 90(14), 2499-2523.

Chen, W., Chun, K.-J., and Barber, R. F. (2018). *Discretized conformal prediction for efficient distribution-free inference.* Stat, 7(1):e173.